# California's zero-emission vehicle adoption brings air quality benefits yet equity gaps persist

Qiao Yu [1], Brian Yueshuai He [2], Jiaqi Ma [2] & Yifang Zhu [1] ✉

Zero-emission vehicle (ZEV) adoption is a key climate mitigation tool, but its environmental justice implications remain unclear. Here, we quantify ZEV adoption at the census tract level in California from 2015 to 2020 and project it to 2035 when all new passenger vehicles sold are expected to be ZEVs. We then apply an integrated traffic model together with a dispersion model to simulate air quality changes near roads in the Greater Los Angeles. We found that per capita ZEV ownership in non-disadvantaged communities (non-DACs) as defined by the state of California is 3.8 times of that in DACs. Racial and ethnic minorities owned fewer ZEVs regardless of DAC designation. While DAC residents receive 40% more pollutant reduction than non-DACs due to inter-community ZEV trips in 2020, they remain disproportionately exposed to higher levels of traffic-related air pollution. With more ZEVs in 2035, the exposure disparity narrows. However, to further reduce disparities, the focus must include trucks, emphasizing the need for targeted ZEV policies that address persistent pollution burdens among DAC and racial and ethnic minority residents.

In California, the transportation sector contributes to approximately 50% of total greenhouse gas (GHG) emissions and 90% of diesel particulate matter (PM) pollution[1]. Unlike internal combustion engine vehicles (ICEVs), zero-emission vehicles (ZEVs) produce no tailpipe emissions and only generate non-exhaust emissions caused by brake and tire wear. ZEV policy, a practical climate change mitigation tool, is expected to produce health co-benefits by reducing traffic-related air pollution (TRAP) while also reducing GHG emissions[2-6].

The California Air Resources Board (CARB) defines ZEVs as vehicle technologies including battery electric, plug-in hybrid, and hydrogen fuel cell vehicles. California was one of the first government bodies in the world to publish ZEV sales requirements: Executive Order N-79-20 of September 2020 requires all new passenger vehicles sold in California to be ZEVs by 2035[7]. In 2022, the CARB approved the Advanced Clean Cars II rule, which establishes a year-by-year roadmap so that by 2035 100% of new cars and light trucks sold in California will be ZEVs[8]. An increasing number of regions across the globe are following

California in accelerating ZEV adoption: New York State adopted California's ZEV rules in September 2022[9]; the United Kingdom announced a plan for new car and van sales to be fully zero-emitting by 2035[10]; and the European Union also reached an agreement to zero-out tailpipe emissions for both new cars and vans by 2035[11]. The International Energy Agency projects that there will be 200 million electric vehicles (EVs) in 2030 globally, 11 times the stock in 2022[12]. As the fifth largest economy in the world and the largest ZEV market in the United States, California has a total of 635,000 registered ZEVs at the end of 2020, which still only represents approximately 2.2% of the total vehicle fleet[13]. Nevertheless, California is expected to make 9.5 million more ZEV sales by 2035[8].

Despite air quality improvements being made in the past few decades, many metropolitan areas and disadvantaged communities (DACs) in California, as defined by the state government, still experience the worst air quality in the country[14]. Here, we use the Senate Bill (SB) 535 DAC designation[15] since it specifically targets greater air

[1]Department of Environmental Health Sciences, Fielding School of Public Health, University of California, Los Angeles, Los Angeles, CA, USA. [2]Department of Civil and Environmental Engineering, Samueli School of Engineering, University of California, Los Angeles, Los Angeles, CA, USA. ✉e-mail: yifang@ucla.edu

pollution reduction and climate change investments. The SB535 DAC designation is based on CalEnviroScreen, a tool developed by the California Environmental Protection Agency to identify California communities disproportionately affected by pollution while considering health, socioeconomic, and population characteristic data. These data are then computed to derive an overall score, which is later utilized to rank communities and identify DACs. The designation comprises four categories: (1) Census tracts in the top 25% of overall scores in CalEnviroScreen 4.0; (2) Tracts with the highest 5% cumulative pollution burden scores lacking overall scores due to data gaps; (3) Tracts identified as disadvantaged in 2017, regardless of CalEnviroScreen 4.0 scores; (4) Lands under federally recognized tribes' control, with a consultation process for tribal designation requests. Over 9.3 million Californians live in DACs[14]. California also has the highest percentage of population living near-roadways (within 300 meters of a major road)[16]. In addition, many DAC residents are racial and ethnic minorities who are often more susceptible to environmental pollutants due to cumulative vulnerabilities they encounter, including biological (epigenetic expression and preexisting health conditions) and social vulnerabilities (low socioeconomic status, ethnoracial discrimination, and physiological stressors)[17–20]. While the term "racial and ethnic minorities" has been officially used by government agencies such as the U.S. National Institutes of Health[21,22], there have been calls to replace it with 'racially and ethnically minoritized' to recognize the active processes of marginalization and systemic discrimination that these groups experience[23,24]. Moreover, due to historical and ongoing socioeconomic inequities, racial and ethnic minority populations often reside near transportation infrastructure[25–29]. This is further compounded by the fact that the vehicle fleet passing through DACs and non-DACs differs considerably. There is a higher proportion of medium- and heavy-duty trucks and older vehicles that emit more pollutants in DACs, resulting in higher levels of TRAP exposure in these communities[30,31]. Therefore, these communities are disproportionately exposed to higher levels of TRAP and other environmental pollutants, together contributing to health disparities[32–36]. Thus, there is a critical need to assess the distributive equity of ZEVs and associated air quality benefits through the lens of environmental justice. A lack of such knowledge will likely undermine the equity perspective during climate mitigation policy planning and aggravate health disparities among DAC residents.

Previous studies have quantified the ambient air quality and health benefits of ZEV adoption at the national and state levels: a 75% fleet electrification rate in the United States could prevent 3000 PM$_{2.5}$-related premature deaths and bring ~$70 billion in health benefits per year[6]. Another study[37] simulating full electrification of light-duty vehicles and buses in California has demonstrated an average PM$_{2.5}$ reduction of 0.13 μg/m³. More recent research[38] presenting various electrification scenarios has reported reductions in PM$_{2.5}$ concentrations ranging from 0.08 to 0.98 μg/m³. Since traffic emissions often contribute most to ambient air pollution in urban environments[39–41], analyzing near-roadway TRAP exposure at the neighborhood level could be both insightful and indicative from an environmental justice perspective. In addition, epidemiological studies have also found linkages between near-roadway exposure to TRAP and a variety of adverse health outcomes, including birth effects and cardiorespiratory morbidity and mortality[42–47]. Focusing on near-roadway TRAP exposure allows us to investigate how ZEV-related air quality benefits are distributed at the community level.

In this study, we analyzed the historical ZEV adoption trend for DACs and non-DACs in California between 2015 and 2020. We then chose the Greater Los Angeles area, which has the largest ZEV population in the United States[13], and conducted a detailed near-roadway air quality analysis based on ZEV adoption data in 2020 to assess the disparities between DACs and non-DACs. To further analyze the extent to which ZEVs can reduce disparities, we conducted additional analysis based on the projected adoption of ZEVs in 2035. We first create an integrated transportation model that combines ZEV ownership data, household travel demand data, and transportation supply data to predict ZEV trips in terms of electric vehicle miles traveled (eVMT) at the roadway link level. Here, eVMT represents the total miles driven by ZEVs in a given census tract. We then analyze ZEV ownership and eVMT per census tract among different racial and ethnic groups. Finally, we calculate emission changes associated with modeled eVMT at each link and use a dispersion model to project fine particulate matter (PM$_{2.5}$) and nitrogen oxides (NO$_x$) concentrations at the census tract level. These results provide evidence for policy-makers to design future ZEV policies to address environmental justice issues related to disproportionately higher exposure to TRAP found among DAC residents and racial and ethnic minorities.

## Results
### ZEV ownership
The three Lorenz curves in Fig. 1 show how plug-in hybrid vehicles (PHEVs) and battery electric vehicles (BEVs) ownership disparities

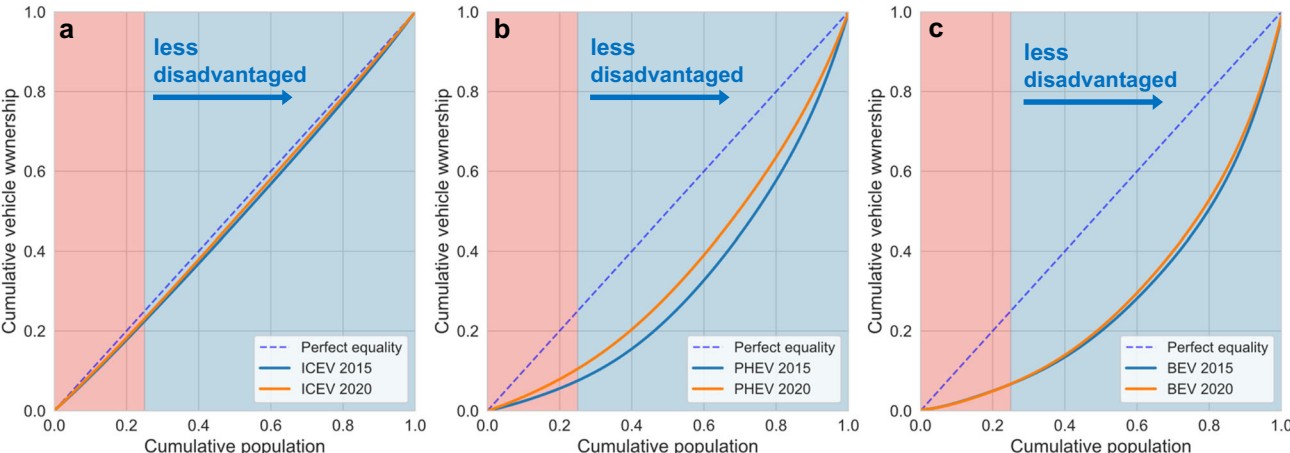

**Fig. 1 | Ownership changes of different vehicle technologies from 2015 to 2020 in California.** Lorenz curves of cumulative vehicle ownership and cumulative population for (**a**) internal combustion engine vehicles (ICEVs), (**b**) plug-in hybrid vehicles (PHEVs), and (**c**) battery electric vehicles (BEVs) from 2015 to 2020 in California. The cumulative population is sorted from highest CalEnviroScreen 4.0 percentiles to lowest percentiles. The red shaded area represents the 25% of the population most disadvantaged, and the blue shaded area represents the remaining 75% of the less disadvantaged population. The blue dashed line in each subfigure represents the ideal Lorenz curve with perfect equality.

changed from 2015 to 2020 relative to ICEVs. The Lorenz curve was initially developed to represent income inequality, where the horizontal axis represents the cumulative population, and the vertical axis represents the cumulative population income[48]. Here, we adapt the Lorenz curve concept and plot cumulative vehicle ownership on the vertical axis. In 2015, the most disadvantaged population (top 25%) shared 23%, 7.6% and 6.6% of all ICEVs, PHEVs, and BEVs, respectively. After 5 years, these numbers are respectively 23%, 11%, and 6.6% for 2020. For ICEVs, the ownership share is almost the same as the cumulative population share based on CalEnviroScreen ranking (Fig. 1a). For PHEVs, the Lorenz curve moves toward the perfect equality line, as ownership disparity has improved despite a large gap remaining: the PHEV share of the most disadvantaged population (top 25%) increases from 7.6% to 11% from 2015 to 2020 (Fig. 1b). For BEVs, the gap is even greater relative to PHEVs, and no change is observed: ownership for the most disadvantaged population (top 25%) stagnates at 6.6% over the five-year period (Fig. 1c).

Figure 2 shows how ZEV ownership changes spatially from 2015 to 2020. Figure 2a, d represent CalEnviroScreen 4.0 percentiles for California and Los Angeles County, respectively. Redder colors reflect more disadvantaged census tracts. In California, from 2015 to 2020,

the average light-duty ZEV ownership per 1000 residents increased from 4.3 to 16 (Fig. 2b, c). However, only a small amount of ZEV ownership increased in DACs, as marked in red in Fig. 2a, especially for areas in northern, central, and southeastern California. Most ZEV ownership increases occur in non-disadvantaged census tracts located in the coastal area of Southern California and the Bay Area. Figure 2d–f shows a similar trend in Los Angeles County: most of the ZEV ownership increase took place in tracts with lower CalEnviroScreen 4.0 percentiles, while the disparity merely improved for tracts located in central Los Angeles County.

## ZEV traffic

Figure 3 compares the spatial patterns between total (Fig. 3a, c) and ZEV traffic (Fig. 3b, d). The ZEV traffic volumes for both freeways and arterial roadways in 2020 (more local traffic activities) are affected by the ZEV ownership data for nearby communities, as presented in Fig. 2f. For freeways, while I-405, one of the major freeways oriented north–south on the left side of Los Angeles County, has a higher percentage of ZEV traffic volume, other highways located in census tracts with lower levels of ZEV ownership have much lower ZEV traffic volumes (Fig. 3b). In addition, I-110, I-710, and CA-60, with more truck

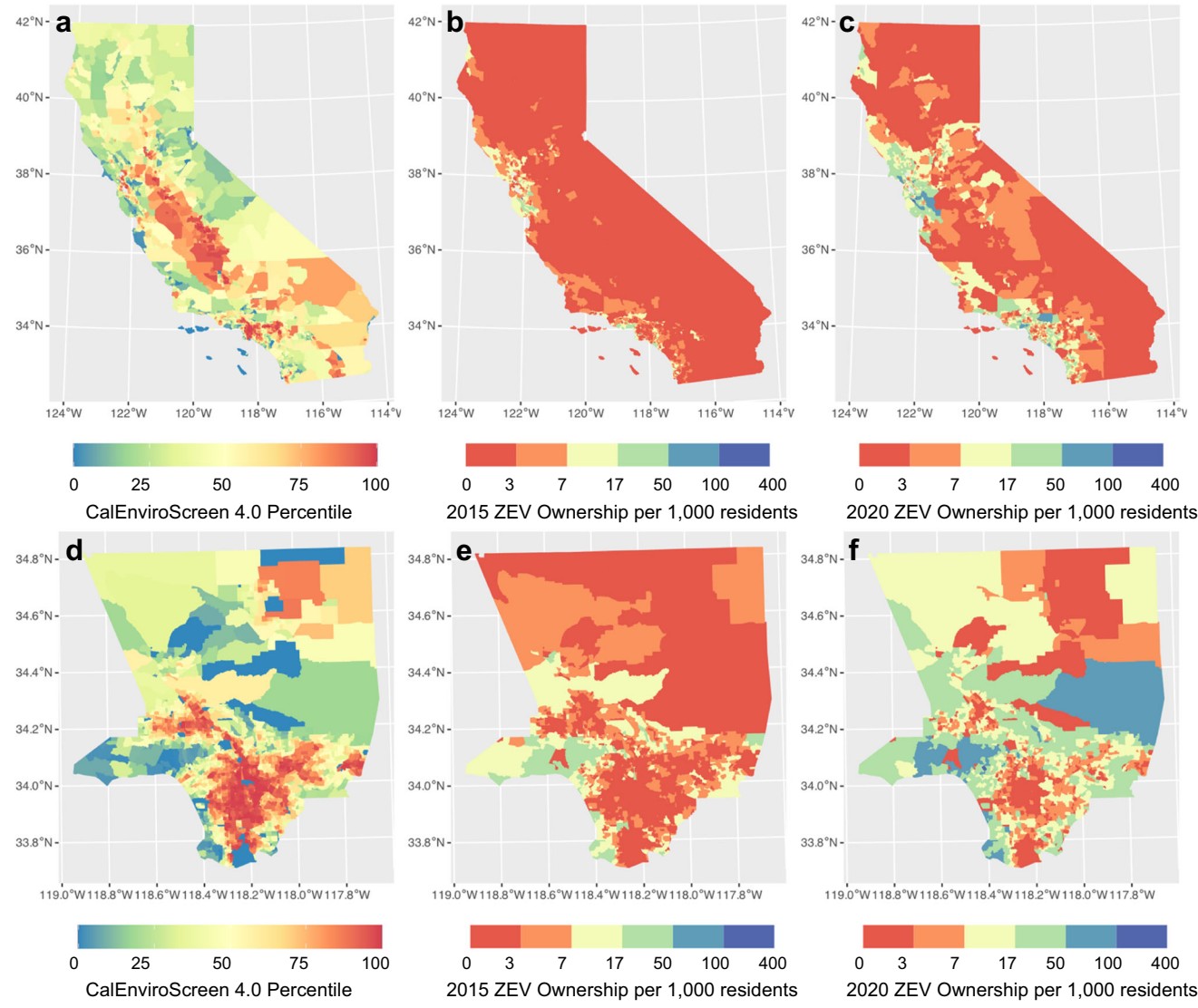

**Fig. 2 | Spatial distribution of zero-emission vehicle (ZEV) ownership from 2015 to 2020 in California and Los Angeles County.** Census tract level spatial distribution for (**a**) the CalEnviroScreen 4.0 percentile, (**b**) 2015, and (**c**) 2020 ZEV ownership per 1,000 residents in California and for (**d**) the CalEnviroScreen 4.0 percentile, (**e**) 2015, and (**f**) 2020 ZEV ownership per 1000 residents in Los Angeles County.

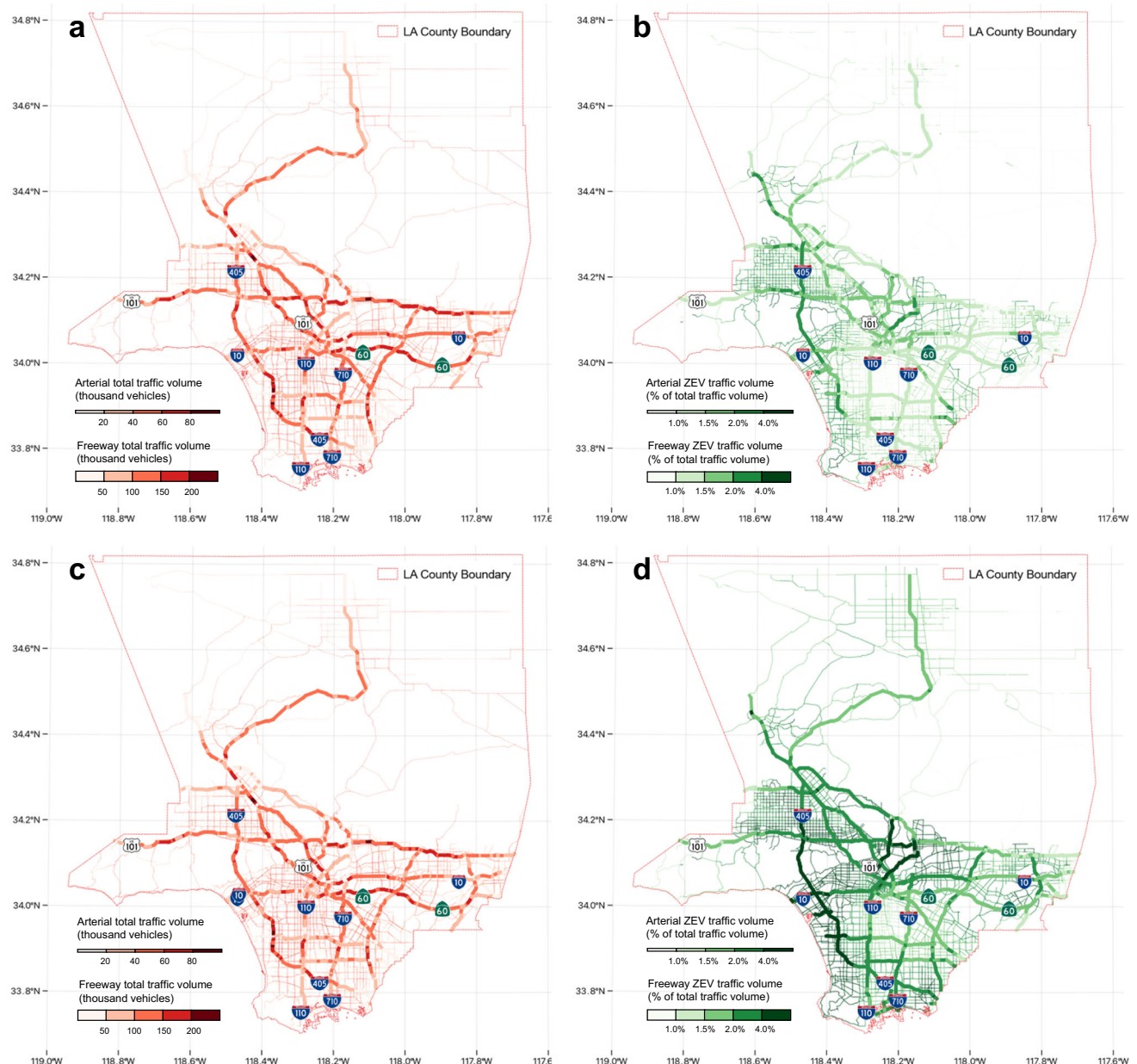

**Fig. 3 | Spatial distribution of simulated daily traffic volumes in Los Angeles County in 2020 and 2035.** Simulated link-level arterial and freeway traffic for (**a**) total traffic volume and (**b**) ZEV traffic volume as a percentage for Los Angeles County in 2020; (**c**) total traffic volume and (**d**) ZEV traffic volume as a percentage for Los Angeles County in 2035.

traffic[49,50], show lower ZEV traffic volumes since medium- and heavy-duty ZEV technology was still less developed in 2020. For arterial roads, links near or located in census tracts with higher levels of ZEV ownership have the highest ZEV traffic volumes. Links located in the lower middle part of Los Angeles County hardly have any ZEV traffic volumes greater than 1%. These links are located in the most disadvantaged communities, which also have the lowest per 1,000 ZEV ownership (Fig. 2d, f). Despite a modest increase in total traffic volume (Fig. 3c) and a spatial pattern similar to that of 2020 (Fig. 3b), there is a notably higher proportion of ZEV traffic observed in 2035 (Fig. 3d) due to increased ZEV adoption.

## Racial and ethnic disparities

Figure 4 shows how ZEV ownership and eVMT which can also be viewed as a surrogate for traffic-attributable emission reduction are distributed among different racial and ethnic groups in Los Angeles County. As shown in Fig. 4a, at the County level, the white population,

while constituting only 26% of the total population, possesses 45% of the ZEV and 31% of the eVMT share. For the Hispanic population, despite showing low ZEV ownership levels of only 26%, the eVMT share is almost doubled at 44%, close to but still lower than its 48% population share. The non-Hispanic Asian American and Pacific Islander population has higher ZEV and eVMT shares than its population share. The non-Hispanic African American population has lower ZEV and eVMT shares than its population share. Populations identified as other races or as multiple races and the Native American population have similar ZEV and eVMT shares relative to their population shares. The ZEV ownership disparity persists regardless of the DAC designation (Fig. 4b, c): the Hispanic or Latino population owns fewer ZEVs relative to its population share even among non-DACs, whereas the non-Hispanic white population residing in DACs still has almost double the ZEV share relative to its population share. The higher eVMT share among Hispanics in DACs can be partially attributed to the more extensive network of highways and roads within their communities,

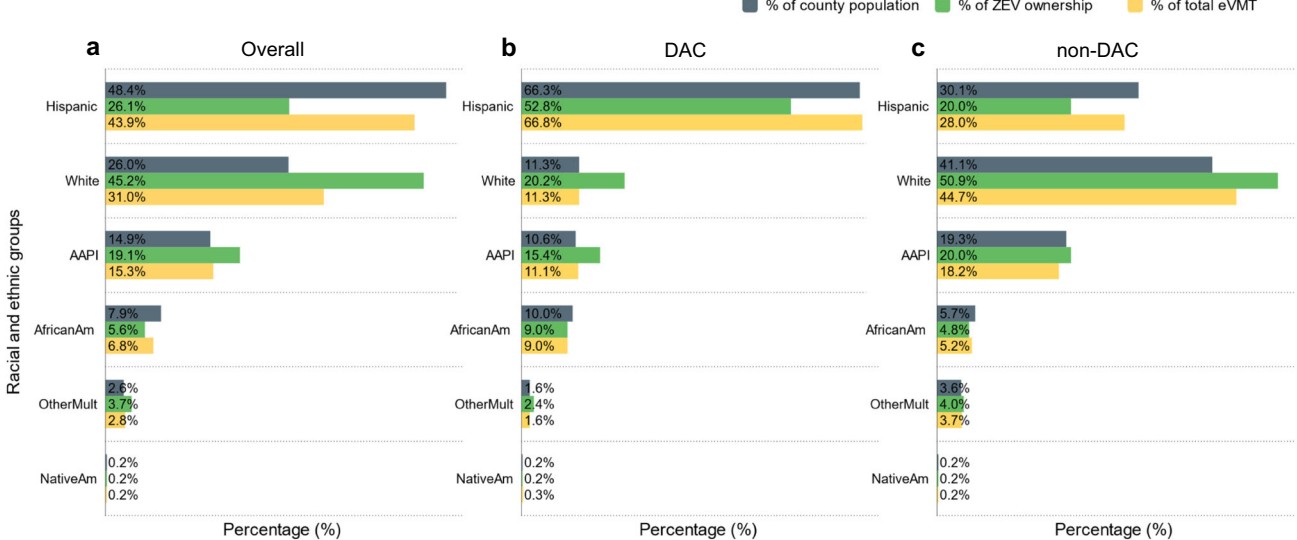

**Fig. 4 | Racial and ethnic analysis on zero-emission vehicle (ZEV) ownership and simulated electric vehicle miles traveled (eVMT) in 2020.** Share of county population, ZEV ownership, and eVMT per census tract for different racial and ethnic groups in (**a**) all communities, (**b**) disadvantaged communities (DACs), and (**c**) non-disadvantaged communities (non-DACs) in Los Angeles County. Hispanic: Hispanic or Latino. White: non-Hispanic white. AAPI: non-Hispanic Asian American and Pacific Islander. AfricanAm: non-Hispanic African American or black. Other-Mult: non-Hispanic "other" or multiple races. NativeAm: non-Hispanic Native American. Racial and ethnic demographic data were obtained from CalEnvir-oScreen 4.0 for each census tract.

which translates to more vehicle usage leading to a higher eVMT share. Racial and ethnic disparities are observed in both the DACs and non-DACs, suggesting that a more targeted ZEV policy should be developed for racial and ethnic minority population to overcome potential barriers such as linguistic isolation.

## Near-roadway air quality

Table 1 compares number of households, ZEV ownership, eVMT, associated emission reduction, and air pollutant levels of DACs and non-DACs within Los Angeles County for the years 2020 and 2035. It should be noted while residents in DACs constitute 25% of the total population state-wide, this number increases to approximately 45% in Los Angeles County. For year 2020, while ZEV ownership is not equally distributed among DACs and non-DACs, with DACs sharing only 18% of all ZEVs, the eVMT disparity is less prominent, indicating that many ZEVs purchased in non-DACs actually travel through DACs.

The baseline concentration levels without ZEVs for traffic-emitted $PM_{2.5}$ and $NO_x$ are doubled in DACs, showing that DACs receive a much higher TRAP burden than non-DACs. The pollutant concentration reduction attributable to ZEVs is small, as the ZEV population in 2020 accounted for only 2.2% of the total vehicle fleet. However, these numbers increase as the ZEV penetration rate increases. By the year 2035, an anticipated increase in the ZEV population from 2.2% to 50% for light-duty vehicles based on CARB Mobile Source Strategy report[51,52], coupled with more stringent emission standards, further reduced traffic-emitted $PM_{2.5}$ and $NO_x$.

As ZEV adoption expands, the gap in traffic-emitted $PM_{2.5}$ concentrations between DACs and non-DACs narrows from 0.22 µg/m³ to 0.18 µg/m³. A recent study[37] simulating full electrification of light-duty vehicles and buses in California has demonstrated an average $PM_{2.5}$ reduction of 0.13 µg/m³. These findings corroborate our results for the 100% ZEV scenario for light-duty vehicles in 2035, which results in an $PM_{2.5}$ reduction of 0.10 µg/m³ for DAC (see Table S1). The reduction could be even greater if medium- and heavy-duty vehicles were also fully converted to ZEVs, as the aforementioned study reported an average $PM_{2.5}$ reduction of 0.24 µg/m³ for a fully zero-emission fleet. Simultaneously, the traffic-emitted $NO_x$ concentration gap between DACs and non-DACs is projected to decrease substantially from

2.6 ppb to 0.88 ppb, attributed to both the prevalence of ZEVs and the more rigorous emission standards enforced by 2035.

Figure 5 illustrates the spatial distribution of $NO_x$ concentrations for the years 2020 and 2035. These results are consistent with those given in Table 1: the $NO_x$ concentration reduction shares for both year 2020 and 2035 are greater in DACs, showing that the widespread adoption of ZEVs could reduce TRAP in DACs. While the $NO_x$ concentration reductions in ppb (Fig. 5a, c) are more evenly distributed regardless of DAC designation, percentage reductions (Fig. 5b, d) are more pronounced in tracts with higher ZEV traffic volumes (Fig. 3b, d). In DAC areas, such as downtown Los Angeles (located centrally at the lower part of the figure), percentage reductions increase from 2020 to 2035 with the rise of the ZEV population. However, these reductions remain relatively low when compared to those in non-DAC areas.

Figure 6 exhibits a similar pattern to Fig. 5, but with a lower magnitude of $PM_{2.5}$ concentration reduction. In other words, even though DACs receive near-roadway air quality benefits from ZEV trips, the current reductions are simply not sufficient for the following reasons: (a) the ZEV-attributable reduction from vehicle start emissions from local trips cannot be shared relative to intercommunity ZEV trips, and (b) medium- and heavy-duty vehicles contribute more to TRAP exposure in DACs and ZEVs are mainly light-duty vehicles. The air quality benefits for $PM_{2.5}$ are also less than those of $NO_x$ reduction, as ZEVs only reduce tailpipe emissions, and brake and tire wear emissions still occur.

## Discussion

This study examines the environmental justice implications of ZEV adoption by analyzing ZEV ownership distributions, simulating ZEV trips, and quantifying changes in near-roadway air pollutants using a bottom-up approach. Even with several targeted ZEV incentive programs and increased incentive allocation to DACs[53], we find that ZEV ownership gaps between DACs and non-DACs still remain, especially for BEVs. We also find a large ZEV ownership gap between white and racial and ethnic minorities, regardless of DAC designation. This echoes findings from recent studies[54,55], which also concluded that racial and ethnic minorities have lower electric vehicle ownership rates, regardless of income. One potential explanation for this disparity could be the purchasing behavior observed among DAC

**Table 1 | Zero-emission vehicle (ZEV) ownership, electric vehicle miles traveled (eVMT), and traffic-emitted air pollutants in disadvantaged communities (DACs) vs. non-DACs in Los Angeles County in 2020 and 2035**

| Variable | 2020 | | 2035 | |
|---|---|---|---|---|
| | DAC (N = 1173) | non-DAC (N = 1167) | DAC (N = 1173) | non-DAC (N = 1167) |
| | Share (%) | | Share (%) | |
| Number of Households | 45% | 55% | 45% | 55% |
| ZEV Ownership | 18% | 82% | 30% | 70% |
| eVMT | 43% | 57% | 46% | 54% |
| Pollutant emission reduction | (tons/year) | | (tons/year) | |
| PM$_{2.5}$ | 0.39 | 0.51 | 11 | 13 |
| NO$_x$ | 6.3 | 8.3 | 56 | 66 |
| CO$_2$ | 16,000 | 21,000 | 500,000 | 590,000 |
| | Geometric Mean (IQR) | | Geometric Mean (IQR) | |
| Traffic-emitted PM$_{2.5}$ concentration (µg/m$^3$)$^a$ | | | | |
| without ZEVs | 0.42 (0.22–0.79) | 0.20 (0.10–0.50) | 0.39 (0.21–0.72) | 0.18 (0.095–0.45) |
| with ZEVs$^b$ | 0.41 (0.22–0.78) | 0.19 (0.10–0.50) | 0.32 (0.17–0.60) | 0.14 (0.075–0.36) |
| reduction attributable to ZEVs | 0.002 (0.001–0.004) | 0.001 (0.001–0.004) | 0.065 (0.034–0.12) | 0.034 (0.017–0.093) |
| Traffic-emitted NO$_x$ concentration (ppb) | | | | |
| without ZEVs | 5.0 (2.6–9.0) | 2.4 (1.2–5.9) | 1.6 (0.87–2.9) | 0.72 (0.37–1.8) |
| with ZEVs$^b$ | 4.9 (2.5–9.0) | 2.3 (1.2–5.9) | 1.1 (0.61–2.1) | 0.45 (0.25–1.1) |
| reduction attributable to ZEVs | 0.09 (0.05–0.18) | 0.06 (0.03–0.16) | 0.47 (0.25–0.85) | 0.25 (0.12–0.70) |

The upper part of the table reports the shares of the number of households, ZEV ownership, simulated eVMT, and corresponding aggregated emission reductions for PM$_{2.5}$, NO$_x$, and CO$_2$ in tons per year for 2020 and 2035. The lower part of the table reports model-simulated pollutant concentrations attributable to traffic for PM$_{2.5}$ and NO$_x$ and the reduction attributable to ZEVs in Los Angeles County SB535 DACs and non-DACs.
$^a$Average annual daily concentration.
$^b$ZEVs accounted for 2.2% of the total light-duty vehicle fleet in 2020, projected to rise to 50% (light-duty vehicle), 16% (medium-duty vehicle), and 20% (heavy-duty vehicle) by 2035.

residents who often lean towards buying used vehicles[56–59]. As of 2020, the market for used ZEVs is somewhat limited, providing fewer opportunities for DAC residents to acquire these vehicles. Additionally, the higher subsidy elasticity of ZEV demand observed among low and medium-income consumers combined with the even greater disparities in rebate allocation could contribute to this discrepancy[55,60]. Furthermore, a recent study[61] highlights that DAC residents encounter disparities in access to EV charging stations in California, which could further contribute to the lower ZEV penetration rate.

Despite the ZEV ownership disparities observed, we found that ZEVs travel across different communities. Thus, even communities with low ZEV ownership levels receive near-roadway air quality benefits from ZEV trips (Fig. 3). We find that DACs receive more near-roadway air quality benefits for NO$_x$ and PM$_{2.5}$ than non-DACs in 2020, and even more in 2035. This finding is encouraging in that ZEVs can offer near-roadway air quality benefits to various communities. Unlike previous research[62] that focused on regional air quality benefits from ZEVs using a top-down approach—based on vehicle registration locations—our study employs a bottom-up methodology centered on ZEV trip routes. Consequently, we discovered that near-roadway air quality benefits can be distributed irrespective of DAC designation, an insight not addressed in the literature.

Nevertheless, when we compare the reduction to the baseline pollutant concentrations in DACs and non-DACs in 2020, we find that the gap persists: DACs receive 40% and 31% fewer relative air quality benefits for NO$_x$ and PM$_{2.5}$, respectively, compared to non-DACs. In other words, although DACs may experience greater reductions in pollutant concentration in terms of µg/m$^3$ or ppb, because the baseline concentrations in DACs are higher, their relative reductions, when considering the higher starting point, are much smaller compared to

those of non-DACs. This could be explained by the fact that ZEVs not only reduce vehicle running emissions (for which the benefits can be shared across different communities) but also reduce vehicle start emissions. In addition, DAC residents are exposed more to TRAP from medium- and heavy-duty trucks, while the current ZEVs are mainly light-duty vehicles. With more ZEVs on-road in the entire fleet, the disparity narrows in our 2035 scenario: DACs only receive 21% and 15% fewer relative air quality benefits for NO$_x$ and PM$_{2.5}$, respectively. While this decline in disparity is promising near roadways, regional air quality and secondary pollutants such as ozone will require more attention in the future. Owing to the complex nature of ozone[63], ZEV adoption, which reduces NO$_x$ emissions, could paradoxically increase ozone concentrations, especially in VOC-limited regions such as Los Angeles County. This has been reported both in the LA100 study[64] conducted by the National Renewable Energy Laboratory and another recent study[37]. Future strategies will need to consider ozone concentrations in a region-specific context.

Similar to our findings, a previous study has shown that Hispanic and African-American populations are much more likely to live in counties with the worst air quality in the United States[65]. A recent study focused on Los Angeles County delivers a similar message: white residents drive more and traverse communities with a higher proportion of racial and ethnic minority populations[25]. Consequently, this results in disproportionately high TRAP burdens on these racial and ethnic minority residents. This agrees with the results of our study: the racial and ethnic minority populations share more eVMT, which are primarily driven by the ZEV trips initiated by white population. Historically, phenomena such as "white flight", in which white families moved from cities to suburbs to avoid increasing diversity, and the highway construction boom of the 1960s, which connected urban and suburban areas, shaped commuting patterns substantially[27–29]. As a result, non-DAC/white populations in metropolitan regions, including Los Angeles, have contributed substantially to commuting emissions due to their tendency to travel more. Beyond exposure, it's important to recognize the cumulative impact where socio-economic, environmental, and health-related factors converge, increasing the susceptibility of DAC residents, especially the racial and ethnic minority populations, to the adverse effects of TRAP. While our primary focus is on exposure, addressing the underlying determinants of this increased susceptibility can amplify the benefits of reducing TRAP exposure. To comprehensively evaluate the synergy between reducing exposure and susceptibility, future studies specifically focusing on susceptibility within DAC and racial and ethnic minority populations are warranted. In addition to the distributive equity issue, DAC residents, especially the racial and ethnic minority populations, are often excluded from collective decision-making processes in contemporary history[66,67]. As a result, unjust redlining investment practices, transportation policies, and urban planning force DAC residents to host unwanted transportation structures in their communities and disproportionately bear higher exposure to TRAP[68]. Policy-makers should also focus on capacity building and procedural equity when developing future targeted ZEV policies.

Different climate mitigation policies could potentially lead to spatial heterogeneities of ambient air quality across communities[69,70]. In terms of spatial resolution, each level of granularity, from near-road to the regional, provides distinct insights and implications. For more accurate environmental justice analyses, considering that individual communities have unique characteristics, it is beneficial to use the finest resolution possible. However, this granularity often requires extensive data collection and computational resources, which might not always be readily available. Thus, adjustments tailored to specific research projects may be needed.

While regional air quality improvements from ZEVs using chemical transport model (CTM) are well studied, a community-level ZEV-related air quality benefit analysis framework has not yet been

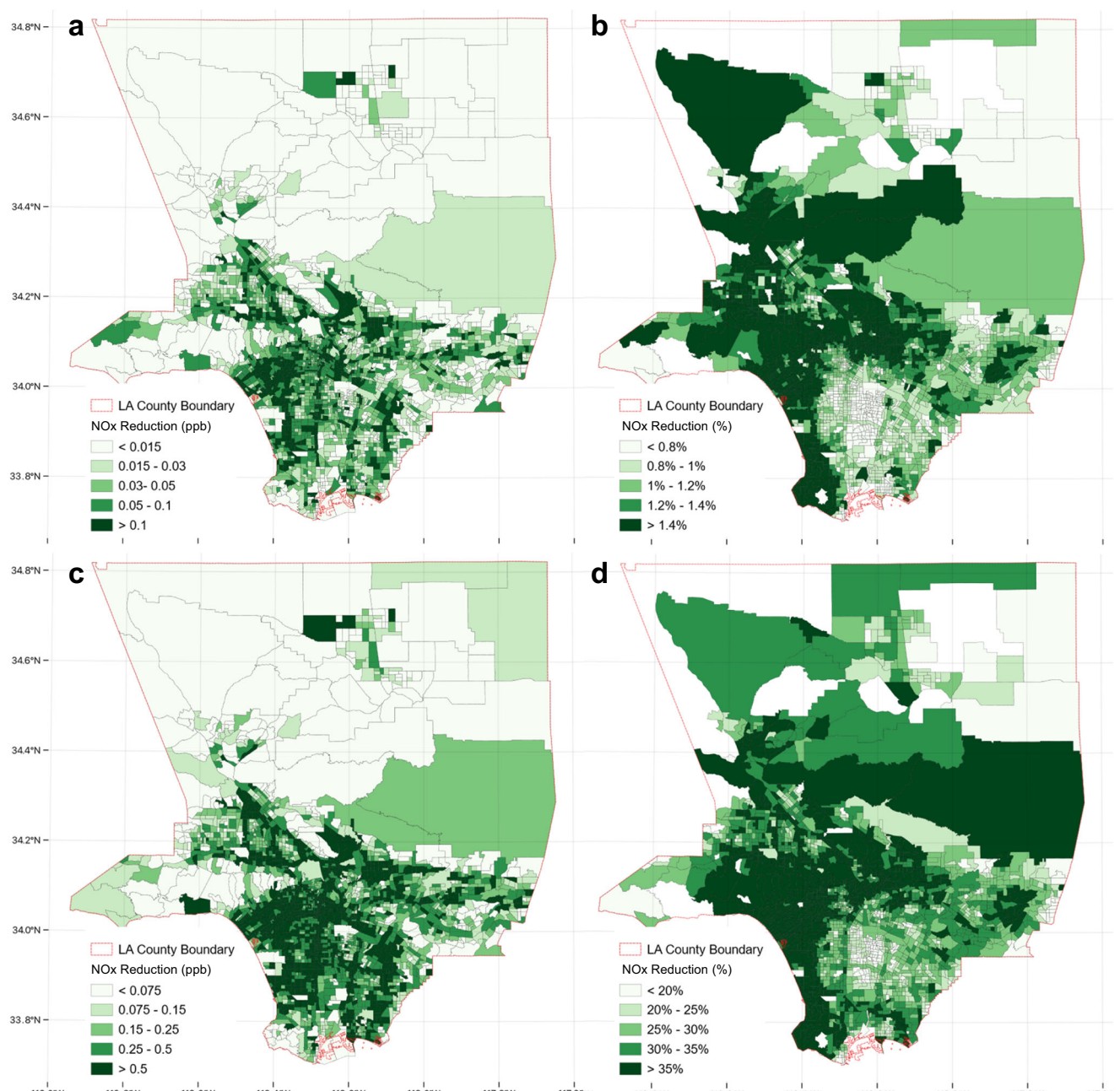

**Fig. 5 | Spatial distribution of near-roadway nitrogen oxides (NO$_x$) reduction attributable to zero-emission vehicles (ZEVs) in Los Angeles County in 2020 and 2035. a** NO$_x$ in ppb, **b** NO$_x$ as a percentage in 2020; **c** NO$_x$ in ppb, **d** NO$_x$ as a percentage in 2035. The percentage (relative reduction) is calculated by dividing the ZEV-attributable reduction by the total traffic-attributable pollutant concentration in each census tract. The deeper the green color, the greater the reduction observed.

established. Our work begins this process and can be further developed into a more comprehensive analysis framework by integrating high-resolution data from other sources (e.g., commercial and residential building emissions) to address environmental justice issues related to climate change and air pollution. Previously, a study[71] reported an association between zip code level ZEV adoption and lower ambient nitrogen dioxide concentrations albeit not statistically significant. By tracking individual trip and utilizing link-level emission data specifically from traffic, our approaches allow us to model the near-roadway air quality benefits attributable to ZEVs, finding statistically significant differences. Our methodology and findings are generalizable to other metropolitan areas, especially those with urban planning histories and demographic distributions similar to those of Los Angeles County. Our results suggest that

ZEVs could bring cross-community near-roadway air quality benefits, yet the air quality disparity persists between DACs and non-DACs at present.

To reduce this disparity, it is critical to ensure a just transition to clean transportation[72,73]. As shown in our 2035 simulation results, the disparity can be reduced with more light-duty ZEVs. Although a universal ZEV incentive program can boost ZEV adoption and benefit DACs, targeted policies are needed to reduce the TRAP exposure gap between DACs and non-DACs, a result of historically unjust land-use policies. Recognizing and rectifying these historical injustices is a cornerstone of a just transition. This can be achieved by directing more rebates and incentives towards disadvantaged communities, providing them with opportunities to access clean transportation.

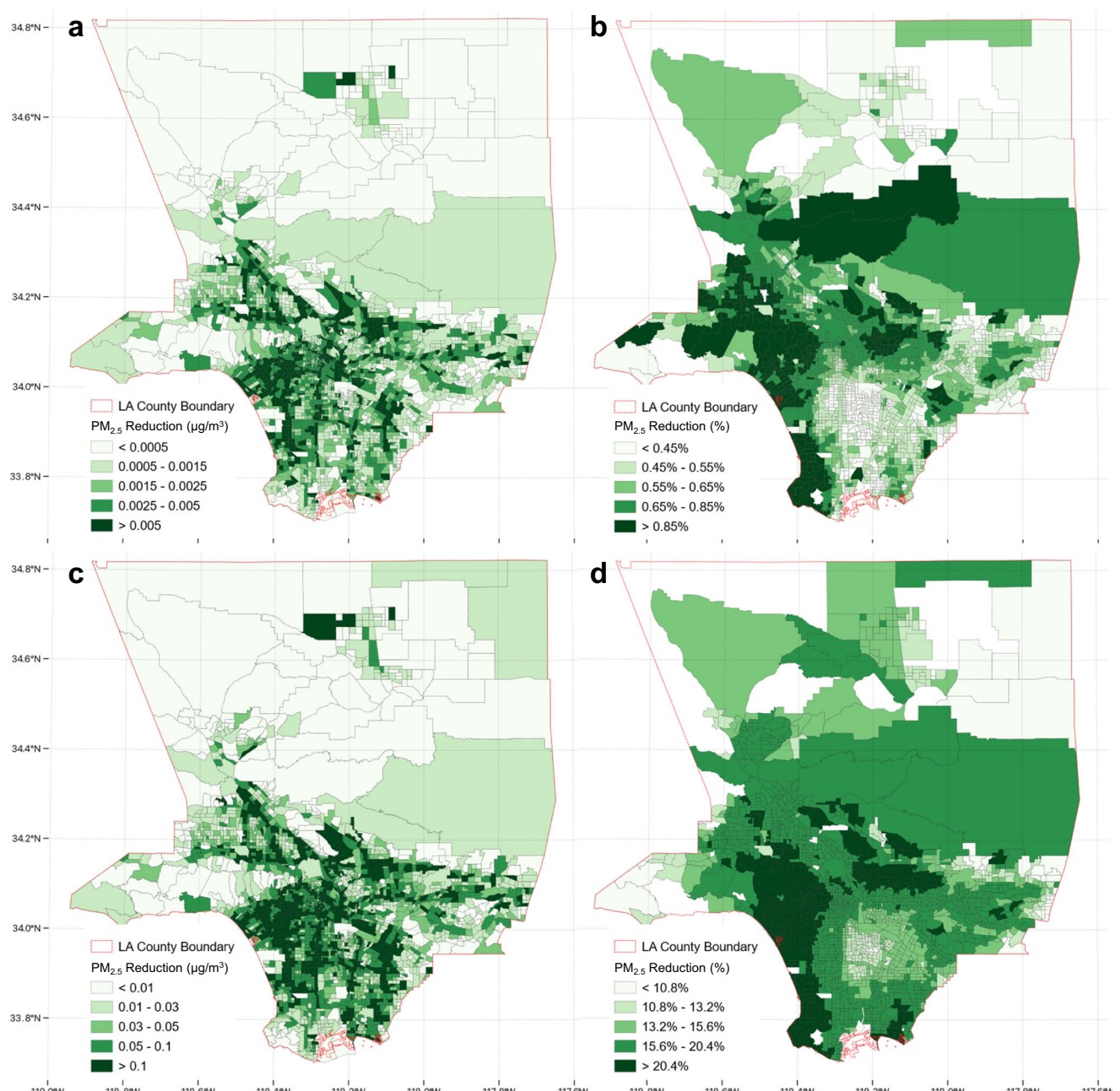

**Fig. 6 | Spatial distribution of near-roadway fine particulate matter (PM$_{2.5}$) reduction attributable to zero-emission vehicles (ZEVs) in Los Angeles County in 2020 and 2035. a** PM$_{2.5}$ in µg/m$^3$, (**b**) PM$_{2.5}$ as a percentage in 2020; (**c**) PM$_{2.5}$ in µg/m$^3$, (**d**) PM$_{2.5}$ as a percentage in 2035. The percentage (relative reduction) is calculated by dividing the ZEV-attributable reduction by the total traffic-attributable pollutant concentration in each census tract. The deeper the green color, the greater the reduction observed.

Addressing the disparities inherent in environmental and health outcomes requires persistent and targeted efforts. As we move forward, future policies and incentive programs should take a comprehensive approach. Trucks, given their substantial emissions and frequent routes through DACs, pose a considerable health risk. It is therefore important to prioritize the transition of trucks to zero-emission alternatives. In addition, addressing non-tailpipe emissions can provide transformative air quality improvements for the most vulnerable communities[69,74,75]. By adopting this holistic approach, we are taking a decisive step towards achieving ZEV distributive justice and ensuring a just transition to clean transportation.

## Methods
### DAC designation
SB 535 DAC designation is based on CalEnviroScreen 4.0, a tool developed by the California Environmental Protection Agency to identify California communities that are disproportionately affected by pollution. CalEnviroScreen 4.0 assigns scores of different indicators to each census tract, and a final score is calculated by multiplying the pollution burden and population characteristics[14]. The final score is then expressed as a percentile ranking. The higher the percentile, the more risk is faced by the census tract residents. Compared to CalEnviroScreen 4.0, the final SB 535 DAC designation includes more DACs from previous CalEnviroScreen versions, recognized Tribes and tracts

with high pollution burden scores but without population characteristics scores. Using SB 535 DAC designation allows us to provide more policy-relevant and readily applicable analysis.

## ZEV adoption trends

To better understand light-duty ZEV adoption trends in different California communities from 2015 to 2020, we retrieve vehicle registration data from the CARB Fleet Database[76] which provides Census Block Group level vehicle population estimation based on registration data from the California Department of Motor Vehicles. Census block group data are then aggregated at the census tract level to facilitate environmental justice analysis.

To project the future ZEV adoption in 2035 when all new passenger vehicles sold are expected to be ZEVs, we applied different logistic growth models to estimate the number of light-duty ZEVs for each census tract within Los Angeles County based on census tract specific historical ZEV adoption data between 2015 to 2020. For medium- and heavy-duty vehicles, we used the ZEV penetration prediction directly from EMFAC2021 v1.0.2[77], an official emission and fleet inventory database developed by the CARB. The logistic growth model is a mathematical framework commonly used to predict the adoption rate of new technologies. It describes a sigmoidal, or S-shaped, curve, representing a slow initial adoption, followed by rapid growth as the technology becomes more prevalent, and eventually leveling off as the market becomes saturated. This model has been widely applied in various fields, including technology diffusion, population growth, and resource consumption, to forecast future trends and inform decision-making processes. We utilized light-duty ZEVs ownership data for each census tract from 2015 to 2020 to establish a growth trend from EMFAC2021 v1.0.2[77]. After estimating the number of light-duty ZEVs for each census tract in 2035, we adjust the total light-duty ZEVs count in Los Angeles County to achieve a final ZEVs penetration rate of 50%, in accordance with the value used in the CARB Mobile Source Strategy report[51,52,78]. We then proportionally scale the ZEVs count for each census tract to reflect this target penetration rate.

To represent this methodology mathematically, we use the following logistic growth equation:

$$N(t) = \frac{K}{1 + \frac{K - N0}{N0} e^{-rt}} \tag{1}$$

where:

N(t) is the number of ZEVs at time t (in our case, t = 2035),

K is the carrying capacity, representing the total light-duty vehicle population being zero-emissions in 2050,

N0 is the initial number of ZEVs (at t = 2015),

r is the growth rate, estimated from the ZEVs ownership data from 2015 to 2020,

t is the time (in years) since the initial year.

The ZEVs fleet penetration rates at the census tract level in 2035 were subsequently utilized in both the transportation and air quality models to provide a comprehensive analysis of near-roadway air quality impacts of ZEVs adoption.

## Integrated transportation model

To estimate the distribution of eVMT, we adopted an integrated transportation model to predict the traffic volume distribution on a typical weekday. The integrated model employs agent-based simulation to simulate the dynamic interactions between travel demand and supply and predict the equilibrium state of the transportation system by leveraging synthetic and real data from multiple sources. Based on the ZEV adoption estimation, ZEV users can be selected from the population, and the associated eVMT distribution can be obtained.

Travel demand is derived from an activity-based model (ABM) developed by the Southern California Association of Government (SCAG) for both 2020 and 2035 scenarios, which is one of the largest models used in practice in the United States. Based on the principle that people's demand for travel is derived from the demand for activity[79], the ABM predicts people's decision-making regarding a series of interdependent travel-related choices and estimates the derived travel demand at the individual level in consideration of spatial-temporal constraints[80,81]. While the ABM predicts the travel demand in six counties of Southern California, we filtered trips within Los Angeles County in this paper. For trips occurring inside or outside Los Angeles County, we aggregated the origins or destinations to the nearest zones on the border of Los Angeles County. Approximately 3,221,000 households (9,661,000 population) in Los Angeles County were incorporated with 43,850,000 trips in total. Additionally, the heavy-duty truck trips were also incorporated as part of the travel demand, including about 368,000 daily truck trips in Los Angeles County.

The ABM provides a realistic prediction of people's travel demand on a typical weekday, while the movement of travelers and vehicles in a transportation network is not explicitly captured. We adopted agent-based simulation toolkit Multi-Agent Transport Simulation v13 (MATSim)[82] to simulate the movement of travelers and vehicles in a multimodal network of Los Angeles County. In the simulation, we used a "Passenger Car Equivalent" of 3.5, indicating that a truck impacts traffic flow equivalently to 3.5 conventional cars[83]. The traffic volume distribution across Los Angeles County can be obtained from the simulation results at a link-level spatial resolution and temporal resolution for each hour. The multimodal network consists of a road network and a transit network including approximately 354,000 links that are incorporated into the multimodal network. The road network is generated from Open-StreetMap data[84], while the transit network is developed from General Transit Feed Specification (GTFS) data[85].

Considering the computational efficiency of large-scale simulation, 10% of the population was simulated in this paper. An iterative calibration approach was adopted to accommodate the road capacity to the 10% population sample[86]. By selecting traffic count data from major freeways across Los Angeles County as the validation set, the simulated traffic volumes were compared to the validation set (Fig. S1). Good agreements were obtained between the simulated and real traffic volumes in Los Angeles County.

## ZEV trip assignment and emission calculation

We calculate emissions in Los Angeles County for both 2020 and 2035, with and without ZEVs. For our 2020 estimate, we use real-world ZEV ownership data from CARB to determine traffic emissions. For 2035, we rely on the projected light-duty ZEV ownership data, which are anticipated to rise to 50% for light-duty vehicles, while medium-duty vehicles and heavy-duty vehicles are projected by the CARB META tool[78] to reach 16% and 20%, respectively. We assume that both ICEVs and ZEVs have the same vehicle miles traveled (VMT), indicating similar driving behaviors and patterns for both vehicle types. To establish a baseline for our study, we consider all on-road light-duty vehicles to be ICEVs, thus excluding ZEVs. Based on the ZEV ownership percentage in a specific census tract, we assume that the same percentage of trips originating from the census tract will be ZEV trips. The ZEV trips are randomly selected from all trips originating from a census tract.

We then aggregate link-level hourly emission rates for $PM_{2.5}$ and $NO_X$. On-road emission rates for Los Angeles County are retrieved from EMFAC2021 v1.0.2[77]. Vehicle category-specific emission rates from EMFAC are matched with vehicle types in the MATSIM model by vehicle weight class (Table S2). Unlike EMFAC, which uses a more detailed vehicle category classification, the MATSIM model only classifies vehicles into four vehicle weight-based categories. Thus, we calculate MATSIM-weighted emission rates from EMFAC using the

equation below:

$$ER\_MATSIM_j^i = \sum ER\_EMFAC_k^i \times VP^k \qquad (2)$$

where $ER\_MATSIM_j^i$ stands for the emission rate of pollutant i for MATSIM weight class j, $ER\_EMFAC_k^i$ stands for the emission rate of pollutant i for EMFAC vehicle category k that falls into MATSIM weight class j, and $VP^k$ stands for the vehicle population proportion of EMFAC vehicle category k with regard to the total vehicle population that falls into MATSIM weight class j.

Emission rates are then matched with link-level hourly vehicle volumes and vehicle activities (starting or stopping a vehicle) to calculate emissions from different emission processes, including running exhaust emissions, start exhaust tailpipe emissions, idling emissions, and brake and tire wear emissions. The emissions from all processes are then aggregated together to reflect the total emissions of a specific link.

## Near-roadway air pollution modeling

Air dispersion models are often used to evaluate TRAP exposure and associated health outcomes among near-roadway communities[87–89]. Here, we use the R-LINE V1.2 model to calculate hourly $PM_{2.5}$ and $NO_x$ concentrations due to on-road vehicle emissions. R-LINE is a line-source dispersion model developed by the U.S. Environmental Protection Agency (EPA) using steady-state Gaussian formulation[90]. It is specifically designed to take in mobile source emissions, which is in line with the link-level emission data we simulated from MATSIM.

R-LINE V1.2 takes in four input files: (a) run parameters, (b) a source file, (c) a receptor file, and (d) a surface meteorology file. We prepare these input files for each link segment simulated in the MATSIM model individually. For run parameters, we use default parameters except the lane width, which is changed to 3.75 m to match the MATSIM model parameter. For the source file, we convert link-specific parameters such as lane numbers and starting and ending coordinates into an R-LINE compatible format. For the receptor file, we create a receptor network for Los Angeles County including 6,423 receptors, each presenting the centroid of a census block group. We select the receptors within 1,500 meters for each source (each link segment in MATSIM) for the receptor file, as several studies show that 1,500 meters is the maximum length at which $PM_{2.5}$ and $NO_x$ directly emitted from traffic cannot be detected[91–93]. Finally, for the surface meteorology file, we first collect hourly surface meteorological data for the area near Los Angeles International Airport from the National Centers for Environmental Information in the Integrated Surface Dataset format[94]. Data for January, April, July and October (a total of 123 days) are collected to represent seasonal variations across the four seasons. We also collect upper air sounding data from the National Oceanic and Atmospheric Administration Radiosonde Database for the area near Los Angeles International Airport[95]. Surface meteorology and upper air sounding data are then processed using AERMET v22112[96], a meteorological data preprocessor provided by the EPA, to generate R-LINE compatible meteorology files.

With the input files, R-LINE V1.2 generates $PM_{2.5}$ and $NO_x$ concentrations from each link segment to all related receptors at an hourly resolution. Since the receptors are of the Census Block Group level, we aggregate the concentrations to the census tract level to facilitate DAC-related analysis. The final concentrations at each receptor are then aggregated to daily concentrations. The seasonal and yearly daily average concentrations are then calculated. We use the average annual daily average concentrations for our final analysis. Additional data processing, analyses, and geospatial visualization were performed using Python 3.9.13[97], along with the Python modules Pandas 1.5.2[98], NumPy 1.23.5[99], SciPy 1.8.1[100], GeoPandas 0.13.2[101], and QGIS 3.26.2[102]. The census tract administrative boundary shapefile used in this study is from CalEnviroScreen 4.0[14].

## Limitations

Our analysis is subject to several limitations. First, we were unable to predict individual ZEV trip precisely. The prediction of ZEV trips requires additional survey data encompassing household-level choices and distinct driving patterns[103,104]. This includes preferences related to ZEV purchase, the availability of private and public EV charging stations, information on the distribution of ZEV incentives, and actual ZEV trip data that could be used to validate our model. Future research is warranted to address these aspects to improve the accuracy of ZEV-trip simulations. Due to the abovementioned data gaps, ZEV trips were randomly selected from all trips originating from a census tract. To ensure that our random selection did not impact our analysis of pollution concertation reduction distribution in DACs and non-DACs, we repeated the entire simulation process four times. Pearson correlation coefficients of the dispersion model results were calculated using Python module SciPy 1.8.1[100] for each census tract between the first random ZEV trip selection results and the additional random selection results[105]. For $PM_{2.5}$ concentration reduction in each census tract, the average correlation coefficient is 0.99 for DACs and 0.98 for non-DACs. For NOx concentration reduction, the average correlation coefficient is 0.95 for DACs and 0.90 for non-DACs. The correlation coefficients indicate that the results obtained from the four additionally simulated results using different random selections are highly correlated with the first. This could be because resident travel patterns in the same census tract share certain homogeneities. Thus, we are confident that our conclusion is not affected by our random selection process.

Second, we assume that ZEVs and ICEVs have identical VMTs. Although we recognize potential variations in driving patterns between ZEV and ICEV drivers, current empirical data yield inconsistent conclusions, making it difficult to adjust our model. The average annual VMT for ICEV is between 11k and 12k miles, while for ZEVs, it ranges from 6k to 15k miles, depending on survey and modeling methods[103,106]. Moreover, most existing empirical data are aggregated, typically at the annual level, which is inadequate to calibrate our agent-based simulation that requires detailed driving log data. Variations in eVMT might affect absolute pollution reduction values, but the relative shares of eVMT and disparities between DAC and non-DAC remain consistent. Thus, our equity-focused findings will not be affected.

Third, the estimated near-roadway air quality benefits are small for both $PM_{2.5}$ and $NO_x$, as the ZEV fleet penetration rate was only 2.2% in 2020. However, these benefits increase substantially as ZEV fleet expands in 2035. While we confirm that the reductions with and without ZEVs are statistically significant using a paired t-test (p-value < 0.001), and that reductions in both $PM_{2.5}$ and $NO_x$ attributable to ZEVs are significant between DACs and non-DACs (p-value < 0.01), we acknowledge that some uncertainty persists. Thus, we conduct our analysis based on traffic-emitted pollutant concentrations alone instead of adding background pollutant concentrations. Moreover, our work aims to complement existing methodologies such as the CTM. It is important to note that the dispersion model has inherent limitations, including challenges in modeling secondary pollutants.

Fourth, while the utilization of annual average daily concentrations is a scientifically sound approach commonly employed in air pollution research and aligns with EPA's long-term exposure standards, we acknowledge that air pollution is associated with both short-term and long-term health effects. Future studies could enhance our understanding by examining these effects at a more granular temporal scale, leveraging our transportation model, which has temporal resolution down to the hour.

Finally, we acknowledge the impacts of upstream emissions from electricity generating units resulting from increased electricity demand due to ZEVs. However, we anticipate that these upstream emissions would have minimal impact on TRAP near roadways. Nevertheless, studies have suggested that a clean energy portfolio is

the key to reducing upstream emissions and ensuring environmental justice[3,62]. Reducing emissions from electricity generating units could yield air quality benefits, even with the increased energy demand from ZEVs[107,108]. Shifting from fossil fuels to clean energy sources has the potential to substantially reduce the exposure of disadvantaged populations to air pollutants emitted from these units, particularly those living in close proximity[109].

## Reporting summary

Further information on research design is available in the Nature Portfolio Reporting Summary linked to this article.

## Data availability

The datasets analyzed in this study are sourced from publicly available databases as cited within the manuscript. These resources are open-access and can be freely accessed for further research and validation. Specifically, the Environmental Justice Index and DAC designation can be accessed from California government websites (https://oehha.ca.gov/calenviroscreen/report/calenviroscreen-40 and https://oehha.ca.gov/calenviroscreen/sb535). California ZEV registration data and emission rate data can be found in the EMFAC Database (https://arb.ca.gov/emfac/fleet-db and https://arb.ca.gov/emfac/). Meteorological data are available from the National Oceanic and Atmospheric Administration (https://www.ncei.noaa.gov/data/global-hourly/archive/isd/ and https://rucsoundings.noaa.gov/). The R-LINE modeled hourly traffic-attributable $NO_x$ and $PM_{2.5}$ concentrations, both with and without ZEVs for all scenarios, can be downloaded from https://figshare.com/s/a95749be8bb3bb8700a9.

## Code availability

The source code for the R-Line model, a research-grade dispersion modeling tool for near-surface releases, is publicly available at the Community Modeling and Analysis System Center's website (https://www.cmascenter.org/r-line/).

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

## Acknowledgements

This work was supported financially by grants from the California Statewide Transportation Research Program (SB 1) Program under grant number LA2205 awarded to J.M., Center for Excellence on New Mobility and Automated Vehicles Project under Award No.693JJ32350027 to awarded to J.M., Sustainable LA Grand Challenge and the Anthony and Jeanne Pritzker Family Foundation supported Zero Emission Healthy Communities project awarded to Y.Z., the Los Angeles Department of Water and Power funded LA100 Equity Strategies Study under grant number 20225060 awarded to Y.Z., University of California Office of the President Climate Action under grant number R02CP6948 to Y.Z., UCLA Dr. Ursula Mandel Scholarship and UCLA Center For Diverse Leadership in Science Fellowship to Q.Y. The views, opinions, findings, and conclusions or recommendations expressed in this paper are strictly those of the authors. They do not necessarily reflect the views of funding

agencies and/or authors' affiliated institutes. We also thank Dr. Lara Cushing for providing insights on the environmental justice background of the manuscript.

## Author contributions
Q.Y., B.H., J.M., and Y.Z. designed the research. B.H. and J.M. performed the transportation analysis. Q.Y. performed the equity and air quality analysis. Y.Z. supervised the research. Q.Y. and B.H. wrote the manuscript and all authors edited the manuscript.

## Competing interests
The authors declare no competing interests.
