## [Peer Review File · Nature Communications]

California's zero-emission vehicle adoption brings air quality benefits yet equity gaps persistEditorial Note: Parts of this Peer Review File have been redacted as indicated to remove third-party material where no permission to publish could be obtained.

REVIEWER COMMENTS

Reviewer #1 (Remarks to the Author):

This paper focused on an important topic, the environmental justice of zero-emission vehicle adoption in the Los Angeles area. The authors used an integrated traffic model and an air pollution dispersion model to simulate air quality changes near roads after adopting ZEVs in Los Angeles. The authors also compared the effect of ZEV adoption on TRAP change across racial/ethnic groups as well as between DACs and non-DACs. The paper is well-written and is of importance to the literature. I highly appreciate the authors' efforts on this important topic in literature. I have a few suggestions for the authors to consider.

1. Introduction – the authors pointed out that California has a high percentage of the population living in DACs and near roadways, respectively, but didn't connect these two. To my knowledge, most DACs in California are in proximity to highways and major roadways. I suggest the authors add a few sentences in the introduction section to illustrate the connection between the distribution of DACs and roadways.

2. Results – line 164: Fig. 2b should be Fig. 3b.

3. Results – Fig. 4: This plot is a little unclear to me. My understanding is that eVMT should be ZEV miles traveled by an individual. However, it seems to me the eVMT shown in this plot accounts for not only ZEV miles traveled by an individual but also ZEV miles generated by different individuals through the census tract where the individual lives. Could the authors clarify the definition of eVMT?

4. Results – It is not surprising to me that Hispanics have a relatively lower ZEV ownership than whites. It is interesting to see Hispanics always have a higher share of eVMT relative to their share of population and ZEV ownership regardless of DAC designation, and whites have the reversed trend. Do the authors have any explanations for this?

5. Discussion – Although the authors found a bigger gap in ZEV ownership between DACs and non-DACs, they also found minor differences between eVMT and subsequent TRAP reduction between DACs and non-DACs. Also, the large difference in the percentage of TRAP reduction between DACs and non-DACs indicates distinct compositions of vehicle fleets driving through DACs and non-DACs: the majority of TRAP in DACs is from heavy-duty vehicles. It is important to discuss the difference in compositions of vehicle fleets between DACs and non-DACs in the introduction section.

6. Discussion – reference 41 should be updated to Boeing, G., Lu, Y., & Pilgram, C. (2023). Local inequities in the relative production of and exposure to vehicular air pollution in Los Angeles. *Urban Studies*, 0(0). <https://doi.org/10.1177/00420980221145403>

7. Methods – have the authors validated the simulated ZEP trips with any practical dataset?

Reviewer #2 (Remarks to the Author):

This paper analyzes California ZEV adoption at the census tract level from 2015 to 2020 and then simulates air quality changes associated with ZEV adoption. The authors compare both ZEV ownership, usage, and resulting air quality benefits across socio-demographics, specifically by comparing census tracts in designated disadvantaged communities (DACs) to non-DACs. The following are my comments and suggestions about the manuscript.

1. While the introduction provides background and policy context, there is almost no discussion of the relevant literature, not do the authors make their contribution clear.

a. There is an almost shocking omission of very relevant papers. The authors need to go back and re-search the literature to make sure they're citing similar work.

i. Several papers have already examined the distribution of ZEV purchases and incentives across income groups, highlighting many of the disparities pointed out in this manuscript. See, for example, Muehlegger and Rapson (2019), Guo and Kontou (2021), and Hsu and Fingerma (2021).

ii. Holland et al. (2019) does a very thorough analysis examining how benefits of ZEVs are spread

across census blocks, discussing disparities across incomes and races/ethnicities. Note that their analysis is at a higher resolution (census blocks rather than census tracts) and incorporates marginal upstream emissions, painting a much more detailed and thorough picture than this manuscript.

b. Compared to the papers mentioned in (a), what is the major contribution of this manuscript?

2. I find Figure 4 hard to interpret. I think it would be useful to also show population shares of the racial/ethnic groups. Looking at the figure, I was surprised how high the Hispanic ZEV ownership share is, until the text explained the ownership share is lower than the population share. The figure should display all relevant information in a stand-alone manner. I'm still curious as to how the Hispanic ZEV share in DACs compares to the population share within DACs... Also, I think in discussing Figure 4 and VMT shares, it would be useful to discuss commuting distance. My understanding is that many lower income people and likely a higher share of households in DACs face longer commutes. It would be interesting to show or discuss average commuting distance or time for these populations (this data is available at the census tract level in the Census/American Community Survey).

3. In Table 1, please add household share by DAC and non-DAC as a first row, to help better interpret ZEV Ownership and eVMT shares. Also, what are the units of pollutant emissions reductions? Aggregate? Per census tract? Over what unit of time?

4. The authors mention in limitations that "we confirm that the reductions with and without ZEVs are statistically significant using a paired t-test (p value < 0.001)." However, is the reduction attributable to ZEVs statistically significant between DACs and non-DACs (for both PM_{2.5} and NO_x)? Based on the confidence intervals, I would guess not...

5. The authors' interpretation of the policy implications is too simplistic and narrow.

a. The point estimates for pollution reductions in DACs for both PM_{2.5} and NO_x are larger than non-DACs. In my opinion, the absolute reduction is more important than the proportional reduction (which I understand is lower for non-DACs based on their cleaner baseline air quality). Also, given non-linearities in impacts of air pollution, a unit improvement in bad air quality is probably worth a lot more than a unit improvement in good air quality. What does the scientific literature on air pollution say about how we should compare these changes?

b. In the results the authors state "Racial and ethnic disparities [in ZEV ownership and eVMT] are observed in both the DACs and non-DACs, suggesting that a more targeted ZEV policy should be developed for BIPOC populations to overcome potential barriers such as linguistic isolation." Why should equity in ZEV ownership be the policy target? If the goal is to reduce emissions or make greater emissions reductions in DACs that have worse air quality, shouldn't the policy maker goal be to electrify the households with the highest VMT or emissions contributions within the DACs, regardless of origin? From an efficiency perspective, wouldn't it make sense to think about minimizing the policy cost per ZEV adoption, or minimizing the policy cost per unit of pollution reduction resulting from ZEV adoption? If the goal is financial equity, aren't there more efficient and cost-effective redistribution policies than subsidizing ZEVs?

c. Do the results support the existing ZEV incentive structure in California (e.g., substantially higher incentives for lower income households)?

6. The authors assume in their analysis that ZEVs are a perfect substitute for ICEVs (though they do not explicitly state, explain, or justify this assumption, which they should). Again, the authors have failed to examine or cite the relevant literature. Numerous studies have shown that when a household purchases a ZEV, they drive it differently. The results from the simulated air pollution model are therefore likely significantly biased.

a. Most ZEV households own more than one car and can substitute across vehicles. See, for example, Davis (2021).

b. ZEVs tend to be driven fewer miles than ICEVs. See, for example Burlig et al. (2021). Also, the 2017 National Household Travel Survey data show that on average across the US, ICEVs are driven 10,790 miles per year, and BEVs are driven only 7,040 miles per year.

7. In their analysis, the authors use annual daily average concentrations of air pollutants, which I don't think is the best way to capture health impacts. My understanding of the literature that it's exposure that matters more- e.g., having a high dose for half the time and a low dose for half the time is worse than having a medium dose the whole time. Indeed, the EPA's NAAQS are crafted to be inline with epidemiological evidence on pollution exposure. I don't think the authors' approach

captures this. I'd like to see them use an alternative measure- perhaps fraction of time with air pollution above a certain level or in non-compliance with NAAQs.

8. Though the authors briefly mention upstream emissions in the limitations section, this is too important an issue to hand-wave away. The authors claim that changes in upstream emissions are likely to be negligible due to the low ZEV penetration rate, however, I don't see why the same argument wouldn't apply to their main analysis. Furthermore, to the extent the implications of the paper are relevant for future policy recommendations, changes in upstream emissions are only going to become more important in the future as ZEV penetration rates increase. Also, there is usually an environmental justice issue associated with electricity generation, with more disadvantaged populations more likely to be located near power plants. It would be very interesting for the authors to incorporate upstream emissions into their analysis. At a minimum, they should perform some back of the envelope calculations or cite some related literature here to assess how they might impact their findings.

References:

Burlig, Fiona, James Bushnell, David Rapson, and Catherine Wolfram. 2021. "Low energy: Estimating electric vehicle electricity use." In AEA Papers and Proceedings, vol. 111, pp. 430-35.

Davis, Lucas. 2021. "Electric Vehicles in Multi-Vehicle Households." Energy Institute at Haas working paper WP 322, Berkeley, CA. Available at <https://www.haas.berkeley.edu/wp-content/uploads/WP322.pdf>.

Guo, Shuocheng, and Eleftheria Kontou. "Disparities and equity issues in electric vehicles rebate allocation." *Energy Policy* 154 (2021): 112291.

Holland, Stephen P., Erin T. Mansur, Nicholas Z. Muller, and Andrew J. Yates. 2019. "Distributional effects of air pollution from electric vehicle adoption." *Journal of the Association of Environmental and Resource Economists* 6(S1): S65-S94.

Hsu, Chih-Wei, and Kevin Fingerman. 2021. "Public electric vehicle charger access disparities across race and income in California." *Transport Policy* 100: 59-67.

Muehlegger, E., & Rapson, D. 2019. *Understanding the Distributional Impacts of Vehicle Policy: Who Buys New and Used Electric Vehicles?* UC Davis: National Center for Sustainable Transportation. <http://dx.doi.org/10.7922/G21Z42N> Available at <https://escholarship.org/uc/item/1q259456>.

Reviewer #3 (Remarks to the Author):

Synopsis: The authors study the distribution of ZEVs in CA/LA county, simulate their link level traffic patterns, and estimate changes in both emissions and pollutant concentrations near roadways. They find that ZEVs have benefited everyone and that the benefits in DACs have been greater, but not so great as to erase systemic and historical disadvantages.

Recommendation: This is a mostly well-written and well-presented study, however, given that it is based on historical changes of just ~2% ZEV adoption, I would contend that the paper would be immensely more impactful if the authors added forward looking simulations of higher adoption percentages. In particular, since the authors have chosen a framework that presents ZEVs as failing to offset longstanding injustices, it seems ideal/critical to include simulations that demonstrate the magnitude and nature of adoption rates required to level the exposure playing field.

Comments and Criticisms:

- The work presented here is immensely relevant. In fact, something similar was published recently: <https://doi.org/10.1016/j.scitotenv.2023.161761> that should be cited here and incorporated into the discussion.
- A second recent study using a much more sophisticated CTM, is cited by the authors, i.e., Skipper et al. (2023). However, its use in the current manuscript is quite limited and curious. For example, the study is only referenced with regard to PM changes, and it is stated that Skipper et al's 100% ZEV simulation results "match" the 2.2% simulation PM magnitude changes presented here. Can the authors explain how these changes match? Given the greater sophistication of Skipper et al.'s CTM simulations, it would be helpful if the authors were to discuss what is and is not achievable with their modeling set up versus that of Skipper et al. For example, Skipper et al. point out that ZEV adoption increases ozone concentrations, a finding previously reported by Pan et al. (2019) in Houston and Peters et al. (2020) in some CONUS locales (both studies are cited by the authors).
- The authors have chosen to frame the standard by which to judge EVs, as equal exposure to pollutants for all populations. I think a reasonable person would agree. Given this framing, and the conclusion that EVs fail this threshold, the manuscript would be immensely more impactful if the authors were able to demonstrate that this threshold could be attained with EVs alone. Perhaps it cannot, in which case maybe that's not the best metric of success against which to judge EVs? However, if there is a scenario of incentives and fractional adoption that results in exposure equity, these authors seem well placed to identify it. Demonstrating this solution would have much more tangible policy relevance than the current results.
- The omission of a conversation about ozone is likely driven by their air quality model limitations, but since the subject of the manuscript is TRAPs (i.e., traffic-related air pollution) discussion of ozone seems like a relevant discussion point. In particular, the authors posit on Lines 317-320 that their model design is easily generalizable to other locales, but perhaps this is not so true when one wants to assess the holistic air quality impacts of ZEVs, as in the case of ozone, given CONUS-wide differences in VOC to NOx regimes.
- Have the authors considered population-weighting their concentration changes?
- I did not find validation information for the baseline R-Line simulations.
- The authors summarize their emissions considered as: "running exhaust emissions, start exhaust tailpipe emissions, and brake and tire wear emissions" on Lines 418-419. Does this include idling emissions? And is the behavior of vehicles and its effect of emission rates considered in the emissions modeling? For example, the emission profile of a vehicle changes as it sits idling and cools.
- In the discussion of R-Line, simulation of NOx is mentioned, but the paper focuses on NO2. Just to clarify, does R-Line simulate NO2, NOx, or both, and which is reported here?
- On Line 375 it is said: "A truck was equivalent to 3.5 passenger cars." What does this mean?
- On Lines 485-492, the topic of upstream emissions is broached. There are two studies that can support this text: <https://doi.org/10.1016/j.atmosenv.2019.04.003> & <https://doi.org/10.1021/acs.est.6b04868> although it may be important to point out that neither is of a sufficient resolution to assess equitable outcomes, as done in citation 66, albeit with an econometric model.

Response to Reviewers' Comments

We would like to express our gratitude to the three reviewers for their constructive comments, which have proved invaluable in enhancing the manuscript. The major revisions made are as follows:

- We have significantly expanded the discussion on the results and methodology by incorporating findings from previous studies on electric vehicles and environmental justice. To provide a more comprehensive contextualization of our conclusions, we have cited an additional 28 papers in the Introduction and Discussion sections.
- In the Method section, we have provided a more detailed definition of the parameters employed in the model. Furthermore, we have conducted new validation on the R-Line model and simulated future scenarios in 2035, considering higher ZEV penetration rates.
- The Results and Discussion sections now contain an expanded analysis of the policy implications arising from our findings.

All the comments raised by the reviewers have been carefully addressed point-by-point, as outlined below in blue font. Changes made to the manuscript and Supplementary Information are highlighted in yellow in the revised manuscript. Once again, we extend our sincere appreciation to the reviewers for generously volunteering their time and expertise, thereby ensuring scientific rigor within the peer-reviewed literature.

Reviewer #1:

This paper focused on an important topic, the environmental justice of zero-emission vehicle adoption in the Los Angeles area. The authors used an integrated traffic model and an air pollution dispersion model to simulate air quality changes near roads after adopting ZEVs in Los Angeles. The authors also compared the effect of ZEV adoption on TRAP change across racial/ethnic groups as well as between DACs and non-DACs. The paper is well-written and is of importance to the literature. I highly appreciate the authors' efforts on this important topic in literature. I have a few suggestions for the authors to consider.

1. Introduction – the authors pointed out that California has a high percentage of the population living in DACs and near roadways, respectively, but didn't connect these two. To my knowledge, most DACs in California are in proximity to highways and major roadways. I suggest the authors add a few sentences in the introduction section to illustrate the connection between the distribution of DACs and roadways.

Thank you for your suggestion. We agree that the connection between the distribution of DACs and roadways is critical. As such, we have added the following sentence to make this link more explicit in the 3rd paragraph in the Introduction section:

“Moreover, due to historical and ongoing socioeconomic inequities, BIPOC and low-income populations often reside near transportation infrastructure²¹⁻²⁵. This is further compounded by the fact that the vehicle fleet passing through DACs and non-DACs differs significantly. There is a higher proportion of medium- and heavy-duty trucks and older vehicles that emit more pollutants in DACs, resulting in higher levels of TRAP exposure in these communities^{26,27}. Therefore, these communities are disproportionately exposed to higher levels of TRAP and other environmental pollutants, together contributing to health disparities²⁸⁻³².”

2. Results – line 164: Fig. 2b should be Fig. 3b.

Thank you for pointing out the discrepancy. We have now corrected the error and updated "Fig. 2b" to "Fig. 3b" in the revised manuscript.

3. Results – Fig. 4: This plot is a little unclear to me. My understanding is that eVMT should be ZEV miles traveled by an individual. However, it seems to me the eVMT shown in this plot accounts for not only ZEV miles traveled by an individual but also ZEV miles generated by different individuals through the census tract where the individual lives. Could the authors clarify the definition of eVMT?

Thank you for your valuable feedback which allowed us to provide necessary clarity regarding the use of eVMT in our manuscript.

VMT, or Vehicle Miles Traveled, is a measurement unit indicating the total miles driven by all vehicles within a specified region over a certain time period (Wisconsin Department of Transportation, 2009; Texas A&M Transportation Institute, 2016). In our paper, we use the term eVMT to represent Electric Vehicle Miles Traveled, building upon the VMT concept, focusing specifically on the total miles driven by electric vehicles in a given region during a specified time frame. For the purpose of our study, eVMT data were aggregated at each link within a census tract.

The eVMT shown in Fig.4 represents the total ZEV miles traveled within each census tract, not the distance covered per individual (e.g., eVMT per capita). This measurement allows us to assess the overall utilization and potential environmental impact of ZEVs within each census tract, irrespective of whether the ZEVs are owned by residents of the census tract or are simply passing through.

To avoid any ambiguity, we have added the following sentence to the last paragraph in the Introduction section and updated the caption of Fig. 4 in the revised manuscript.

“Here, eVMT represents the total miles driven by ZEVs in a given census tract.”

References:

Wisconsin Department of Transportation (2009). Vehicle miles of travel (VMT). <https://wisconsindot.gov/pages/projects/data-plan/veh-miles/default.aspx>

Texas A&M Transportation Institute (2016). Methodologies Used to Estimate and Forecast Vehicle Miles Traveled (VMT).
<https://static.tti.tamu.edu/tti.tamu.edu/documents/PRC-2016-2.pdf>

4. Results – It is not surprising to me that Hispanics have a relatively lower ZEV ownership than whites. It is interesting to see Hispanics always have a higher share of eVMT relative to their share of population and ZEV ownership regardless of DAC designation, and whites have the reversed trend. Do the authors have any explanations for this?

Thank you for your insightful observations. Indeed, Hispanics, despite having a lower ZEV ownership compared to whites, show a higher share of eVMT relative to their population and ZEV ownership. This pattern is specifically noticeable among Hispanics living in Disadvantaged Communities (DAC). As we previously noted, eVMT represents the total ZEV miles traveled within a census tract which correlates well with the total VMT within the same census tract in our study. The increased eVMT share among Hispanics in DACs thus can be partially attributed to the more extensive network of highways and roads within their communities, which translates to more vehicle usage leading to higher VMT and eVMT shares.

To clarify this point, we have revised Fig. 4 to visually present this trend more clearly and add the following sentence in the Results under the “Racial and ethnic disparities” sub-section.

“The higher eVMT share among Hispanics in DACs can be partially attributed to the more extensive network of highways and roads within their communities, which translates to more vehicle usage leading to a higher eVMT share.”

5. Discussion – Although the authors found a bigger gap in ZEV ownership between DACs and non-DACs, they also found minor differences between eVMT and subsequent TRAP reduction between DACs and non-DACs. Also, the large difference in the percentage of TRAP reduction between DACs and non-DACs indicates distinct compositions of vehicle fleets driving through DACs and non-DACs: the majority of TRAP in DACs is from heavy-duty vehicles. It is important to discuss the difference in compositions of vehicle fleets between DACs and non-DACs in the introduction section.

Thank you for your suggestion to discuss the difference in compositions of vehicle fleets between DACs and non-DACs. We agree that this is an important factor contributing to the observed disparity in TRAP reduction percentages.

Combining your suggestion in comment #1, we have added the following sentences in the 3rd paragraph in the Introduction section of the revised manuscript:

“Moreover, due to historical and ongoing socioeconomic inequities, BIPOC and low-income populations often reside near transportation infrastructure. This is further

compounded by the fact that the vehicle fleet passing through DAC and non-DAC differs significantly. There is a higher proportion of medium- and heavy-duty trucks and older vehicles that emit more pollutants in DAC, resulting in higher levels of TRAP exposure in these communities. Therefore, these communities are disproportionately exposed to higher levels of TRAP and other environmental pollutants, together contributing to health disparities²⁸⁻³².”

Since the revised manuscript now includes ZEV scenarios in 2035 which makes the compositions of vehicle fleets even more important, we also add the following discussion to highlight this point:

Third last paragraph in the Results section:

“The reduction could be even greater if medium- and heavy-duty vehicles were also fully converted to ZEVs, as the aforementioned study reported an average PM_{2.5} reduction of 0.24 µg/m³ for a fully zero-emission fleet.”

Last paragraph in the Discussion section:

“To continue narrowing the gap, future policies and incentive programs should not only focus on DAC residents, but also tackle non-tailpipe emissions and specifically target trucks. These vehicles emit higher levels of pollutants and often travel through DACs”

6. Discussion – reference 41 should be updated to Boeing, G., Lu, Y., & Pilgram, C. (2023). Local inequities in the relative production of and exposure to vehicular air pollution in Los Angeles. *Urban Studies*, 0(0).<https://doi.org/10.1177/00420980221145403>

Thank you for pointing that out. We have now updated the reference 41 as per your suggestion. New reference number is 21.

7. Methods – have the authors validated the simulated ZEV trips with any practical dataset?

Thank you for your inquiry regarding the validation of our simulated ZEV trips.

In our study, we have validated simulated trips for all vehicles with available measurement data as shown in Figure S2 in the original submission, now Figure S3 in the revised manuscript. Unfortunately, at present, there is no practical dataset for ZEV-specific trips that we are aware of. We acknowledge that this is a limitation in our current methodology and have indicated as such in the Method section under the Limitations sub-section of the revised manuscript as shown below.

“The prediction of ZEV trips requires additional survey data encompassing household-level choices. This includes preferences related to ZEV purchase, the availability of private and public EV charging stations, information on the distribution of ZEV incentives, and actual ZEV trip data that could be used to validate our model. Future research is warranted to address these aspects to improve the accuracy of ZEV-trip simulations.”

Reviewer #2:

This paper analyzes California ZEV adoption at the census tract level from 2015 to 2020 and then simulates air quality changes associated with ZEV adoption. The authors compare both ZEV ownership, usage, and resulting air quality benefits across socio-demographics, specifically by comparing census tracts in designated disadvantaged communities (DACs) to non-DACs. The following are my comments and suggestions about the manuscript.

1. While the introduction provides background and policy context, there is almost no discussion of the relevant literature, not do the authors make their contribution clear.
 - a. There is an almost shocking omission of very relevant papers. The authors need to go back and re-search the literature to make sure they're citing similar work.
 - i. Several papers have already examined the distribution of ZEV purchases and incentives across income groups, highlighting many of the disparities pointed out in this manuscript. See, for example, Muehlegger and Rapson (2019), Guo and Kontou (2021), and Hsu and Fingerman (2021).

Thank you for your suggestion to reference these papers. We have reviewed these papers when working on the original manuscript and appreciated their significance. In fact, our original manuscript has referenced the work by Guo and Kontou (2021) (see original reference 39). We did not cite Hsu and Fingerman (2021) and Muehlegger and Rapson (2019) for specific reasons. Hsu and Fingerman (2021) focused primarily on disparities in EV Charging Stations (EVCS). Muehlegger and Rapson (2019) provided data of purchase behaviors for low- and medium-income households. While the two studies are EV-related, they are less relevant in the context of near-roadway air quality impacts of ZEVs, thus we did not cite them in the original manuscript. However, we appreciate your suggestion and agree that a broader discussion of EV-related topics would be helpful in the Introduction section. We thus incorporated the suggested papers along with other relevant studies that we have reviewed together with those recently published since we submitted the original manuscript into the revised manuscript. The added references are listed below:

Canepa, Kathryn, Hardman, Scott, & Tal, Gil. An early look at plug-in electric vehicle adoption in disadvantaged communities in California. *Transp. Policy* 78, 19–30 (2019).

Hennessey, Eleanor M., & Syal, Sita M. Assessing justice in California's transition to electric vehicles. *iScience* 26, 106856 (2023).

DeShazo, J. R., Sheldon, Tamara L., & Carson, Richard T. Designing policy incentives for cleaner technologies: Lessons from California's plug-in electric vehicle rebate program. *J. Environ. Econ. Manage.* 84, 18–43 (2017).

Pierce, Gregory, Deshazo, J R, Sheldon, Tamara, & Blumenberg, Evelyn. Designing Light-Duty Vehicle Incentives for Low- and Moderate-Income Households. (2019).

Muehlegger, Erich, Rapson, David, & Org, Escholarship. Understanding the Distributional Impacts of Vehicle Policy: Who Buys New and Used Electric Vehicles? Publication Date. (2019). doi:10.7922/G21Z42N

Pierce, Gregory, McOmber, Britta, & DeShazo, J.R. Supporting Lower-Income Households' Purchase of Clean Vehicles: Implications From California-Wide Survey Results. (2020).

Muehlegger, Erich, & Rapson, David S. Subsidizing low- and middle-income adoption of electric vehicles: Quasi-experimental evidence from California. *J. Public Econ.* 216, 104752 (2022).

Hsu, Chih Wei, & Fingerman, Kevin. Public electric vehicle charger access disparities across race and income in California. *Transp. Policy* 100, 59–67 (2021).

Chang, Shih Ying, Huang, Jiaoyan, Chaveste, Melissa R., Lurmann, Frederick W., Eisinger, Douglas S., Mukherjee, Anondo D., Erdakos, Garnet B., Alexander, Marcus, & Knipping, Eladio. Electric vehicle fleet penetration helps address inequalities in air quality and improves environmental justice. *Commun. Earth Environ.* 2023 41 4, 1–15 (2023).

ii. Holland et al. (2019) does a very thorough analysis examining how benefits of ZEVs are spread across census blocks, discussing disparities across incomes and races/ethnicities. Note that their analysis is at a higher resolution (census blocks rather than census tracts) and incorporates marginal upstream emissions, painting a much more detailed and thorough picture than this manuscript.

We appreciate the detailed analysis of the distribution effects of air pollution due to electric vehicle adoption presented by Holland et al. (2019). Their work is highly regarded by us as well as by others including reviewer #3. However, Holland et al. (2019) used a top-down approach and conducted an econometric analysis at the national scale. They then used a reduced complexity model to simulate regional air quality at the county level and distributed county-level estimates to each census block based on population distribution.

In contrast, our study uses a bottom-up approach to assess the impacts of ZEVs across different communities. Instead of regional air quality, we focused on changes in near-road air quality due to traffic emissions, utilizing a high spatial resolution of 50 meters. This bottom-up approach allows us to find that many ZEVs purchased in non-DACs actually travel through DACs. Our pollution concentration analysis is conducted at the census block level based on link-level emission data, which is then aggregated to the census tract level to facilitate environmental justice analysis. This is in alignment with the SB535 and CalEnviroScreen 4.0, which use census tracts as their geographic definition unit. Our high-resolution analysis allows us to examine environmental justice perspectives with greater detail and precision, offering a novel contribution to the field.

b. Compared to the papers mentioned in (a), what is the major contribution of this manuscript?

As explained above, the major contributions of this manuscript include:

1. Detailed Bottom-Up Approach: Our study employs a bottom-up approach to analyze changes in near-road air quality due to traffic emissions using a high resolution traffic model, providing a granular perspective for environmental justice analysis.

2. Environmental Justice Focus: Our research emphasizes environmental justice perspectives at the community level, aligning with SB535 and CalEnviroScreen 4.0 guidelines. We investigate the effects of ZEV adoption on TRAP changes across racial/ethnic groups and between DACs and non-DACs.

3. In the revised manuscript, we have included additional 2035 scenarios to further analyze the extent to which ZEVs can reduce the disparities between DACs and non-DACs. These findings provide new insights into the environmental effects of electric vehicle adoption, offering valuable information for policymakers.

2. I find Figure 4 hard to interpret. I think it would be useful to also show population shares of the racial/ethnic groups. Looking at the figure, I was surprised how high the Hispanic ZEV ownership share is, until the text explained the ownership share is lower than the population share. The figure should display all relevant information in a stand-alone manner. I'm still curious as to how the Hispanic ZEV share in DACs compares to the population share within DACs... Also, I think in discussing Figure 4 and VMT shares, it would be useful to discuss commuting distance. My understanding is that many lower income people and likely a higher share of households in DACs face longer commutes. It would be interesting to show or discuss average commuting distance or time for these populations (this data is available at the census tract level in the Census/American Community Survey).

We appreciate the feedback. In the original Figure 4, population share for each racial/ethnic group is given by a vertical blue line denoted as “% of population” in the figure legend. We acknowledge that it might not be as clear as we intended. Therefore, we have revised Figure 4 to improve its clarity and readability as shown below.

Fig. 4: Racial and ethnic analysis on zero-emission vehicle (ZEV) ownership and simulated electric vehicle miles traveled (eVMT) in 2020. Share of county population, ZEV ownership, and eVMT per census tract for different racial and ethnic groups in (a) all communities, (b) disadvantaged communities (DACs), and (c) non-disadvantaged communities (non-DACs) in Los Angeles County. **Hispanic:** Hispanic or Latino. **White:** non-Hispanic white. **AAPI:** non-Hispanic Asian American and Pacific Islander. **AfricanAm:** non-Hispanic African American or black. **OtherMult:** non-Hispanic “other” or multiple races. **NativeAm:** non-Hispanic Native American. Racial and ethnic demographic data were obtained from CalEnviroScreen 4.0 for each census tract.

When discussing commuting time, studies indeed show that it can be longer for low-income individuals, as they may not own personal vehicles, thereby relying on public transit which increases their commute time (Blumenberg et al., 2015). However, commuting distance is a separate consideration. We appreciate your suggestion, but as cited in our manuscript (Boeing et al., 2023), the reality in regions like Los Angeles County is actually the opposite. The residents of non-DAC communities, predominantly white individuals, typically drive more, producing more emissions.

This pattern has historical roots in phenomena such as “white flight”, a trend where white families relocated from cities to suburbs to avoid increasingly diverse or minority-majority neighborhoods (Frey, 1979). Along with this, the highway construction boom in the 1960s, designed to link urban and suburban areas, influenced both settlement and commuting patterns substantially. While we acknowledge that the commuting patterns may vary across different regions, this observed trend in major metropolitan areas suggests that non-DAC/white populations in Los Angeles tend to travel more extensively and therefore contribute more to commuting emissions (Galster, 1990; Frey, 1980). We have added the following sentences in the Discussion section of the revised manuscript to make these points clearer:

“Historically, phenomena such as “white flight”, in which white families moved from cities to suburbs to avoid increasing diversity, and the highway construction boom of the 1960s, which connected urban and suburban areas, shaped commuting patterns

substantially²³⁻²⁵. As a result, non-DAC/white populations in metropolitan regions, including Los Angeles, have contributed significantly to commuting emissions due to their tendency to travel more.”

References:

Blumenberg, Evelyn, Pierce, Gregory, & Smart, Michael. Transportation Access, Residential Location, and Economic Opportunity: Evidence From Two Housing Voucher Experiments. *Citiescape A J. Policy Dev. Res.* 17, 89–111 (2015).

Boeing, G, Lu, Y G, & Pilgram, C. Local inequities in the relative production of and exposure to vehicular air pollution in Los Angeles. *Urban Stud.* (2023).
doi:10.1177/00420980221145403

Frey, W. H. Central city white flight: racial and nonracial causes. *Am. Sociol. Rev.* 44, 415–448 (1979).

Galster, George C. White Flight from Racially Integrated Neighbourhoods in the 1970s: The Cleveland Experience. *Urban Stud.* 27, 385–399 (1990).

Frey, W. H. Black in-migration, white flight, and the changing economic base of the central city. *Am. J. Sociol.* 85, 1396–1417 (1980).

3. In Table 1, please add household share by DAC and non-DAC as a first row, to help better interpret ZEV Ownership and eVMT shares. Also, what are the units of pollutant emissions reductions? Aggregate? Per census tract? Over what unit of time?

Thank you for the suggestions. We have now incorporated the household share by DAC and non-DAC into the first row of Table 1 as shown below to improve interpretability of the ZEV Ownership and eVMT shares.

In terms of pollutant emission reductions, they are presented as total emissions in tons per year. Since our revised manuscript now includes simulation in 2035, we also added emission reduction data in 2035 to Table 1. These emissions reductions are aggregated across all DAC and non-DAC census tracts. We have revised Table 1 caption to clarify these points.

Table 1: Zero-emission vehicle (ZEV) ownership, electric vehicle miles traveled (eVMT), and traffic-emitted air pollutants in disadvantaged communities (DACs) vs. non-DACs in Los Angeles County in 2020 and 2035.

Variable	2020		2035	
	DAC (N=1173)	non-DAC (N = 1167)	DAC (N=1173)	non-DAC (N = 1167)
	Share (%)		Share (%)	
Number of Households	45%	55%	45%	55%
ZEV Ownership	18%	82%	30%	70%
eVMT	43%	57%	46%	54%
Pollutant emission reduction	(tons/year)		(tons/year)	
PM_{2.5}	0.39	0.51	11	13
NO_x	6.3	8.3	56	66
CO₂	16,000	21,000	500,000	590,000
	Geometric Mean (IQR)		Geometric Mean (IQR)	
Traffic-emitted PM_{2.5} concentration (µg/m³)^a				
without ZEVs	0.42 (0.22-0.79)	0.20 (0.10-0.50)	0.39 (0.21-0.72)	0.18 (0.095-0.45)
with ZEVs^b	0.41 (0.22-0.78)	0.19 (0.10-0.50)	0.32 (0.17-0.60)	0.14 (0.075-0.36)
reduction attributable to ZEVs	0.002 (0.001-0.004)	0.001 (0.001-0.004)	0.065 (0.034-0.12)	0.034 (0.017-0.093)
Traffic-emitted NO_x concentration (ppb)				
without ZEVs	5.0 (2.6-9.0)	2.4 (1.2-5.9)	1.6 (0.87-2.9)	0.72 (0.37-1.8)
with ZEVs^b	4.9 (2.5-9.0)	2.3 (1.2-5.9)	1.1 (0.61-2.1)	0.45 (0.25-1.1)
reduction attributable to ZEVs	0.09 (0.05-0.18)	0.06 (0.03-0.16)	0.47 (0.25-0.85)	0.25 (0.12-0.70)

The upper part of the table reports the shares of the number of households, ZEV ownership, simulated eVMT, and corresponding aggregated emission reductions for PM_{2.5}, NO_x, and CO₂ in tons per year for 2020 and 2035. The lower part of the table reports model-simulated pollutant concentrations attributable to traffic for PM_{2.5} and NO_x and the reduction attributable to ZEVs in Los Angeles County SB535 DACs and non-DACs.

^aAverage annual daily concentration

^bZEVs accounted for 2.2% of the total light-duty vehicle fleet in 2020, projected to rise to 50% (light-duty vehicle), 16% (medium-duty vehicle), and 20% (heavy-duty vehicle) by 2035

4. The authors mention in limitations that “we confirm that the reductions with and without ZEVs are statistically significant using a paired t-test (p value < 0.001).” However, is the reduction attributable to ZEVs statistically significant between DACs and non-DACs (for both PM_{2.5} and NO_x)? Based on the confidence intervals, I would guess not...

Thank you for your question. We have conducted two-sample t-tests on the reductions attributable to ZEVs between DACs and non-DACs for both PM_{2.5} and NO_x. In the year 2020, both tests yielded p-values less than 0.01, indicating a statistically significant difference between DACs and non-DACs. Similarly, in the year 2035, the p-values for both pollutants were less than 0.001, further supporting the presence of a statistically significant difference of the reduction attributable to ZEVs between DACs and non-DACs. The following sentences have been added to the Method section to clarify this point.

“While we confirm that the reductions with and without ZEVs are statistically significant using a paired t-test (p-value < 0.001), and that reductions in both PM_{2.5} and NO_x attributable to ZEVs are significant between DACs and non-DACs (p-value < 0.01), we acknowledge that some uncertainty persists.”

5. The authors’ interpretation of the policy implications is too simplistic and narrow.

a. The point estimates for pollution reductions in DACs for both PM_{2.5} and NO_x are larger than non-DACs. In my opinion, the absolute reduction is more important than the proportional reduction (which I understand is lower for non-DACs based on their cleaner baseline air quality). Also, given non-linearities in impacts of air pollution, a unit improvement in bad air quality is probably worth a lot more than a unit improvement in good air quality. What does the scientific literature on air pollution say about how we should compare these changes

We agree with your point about the importance of absolute reductions. Indeed, this aspect is presented in Figures 5 and 6, and we have now further emphasized its significance in the second last paragraph in the Results section:

“While the NO_x concentration reductions in ppb (Figs. 5a, 5c) are more evenly distributed regardless of DAC designation, percentage reductions (Figs. 5b, 5d) are more pronounced in tracts with higher ZEV traffic volumes and fewer medium- and heavy-duty traffic activities (Figs. 3b, 3d). In DAC areas, such as downtown Los Angeles (located centrally at the lower part of the figure), percentage reductions increase from 2020 to 2035 with the rise of the ZEV population. However, these reductions remain relatively low when compared to those in non-DAC areas.

Figure 6 exhibits a similar trend to Figure 5, but with a lower magnitude of PM_{2.5} concentration reduction.”

Regarding non-linearities of air pollution health effects, two key publications—one from the New England Journal of Medicine (Liu et al., 2019) and another from the Proceedings of the National Academy of Sciences (Burnett et al., 2018)—have studied the short-term (daily) and long-term (annual) effects. Both publications found the dose-response curve to be steeper at lower pollutant exposure levels (see figures below). This suggests that while improving air quality in areas with high pollution levels is critical, further improving air quality in relatively cleaner areas is also important.

[redacted]

(Figure reproduced from Liu et al., 2019 and Burnett et al., 2018)

References:

Liu, Cong, Chen, Renjie, Sera, Francesco, Vicedo-Cabrera, Ana M., Guo, Yuming, Tong, Shilu, Coelho, Micheline S.Z.S., Saldiva, Paulo H.N., Lavigne, Eric, Matus, Patricia, Valdes Ortega, Nicolas, Osorio Garcia, Samuel, Pascal, Mathilde, Stafoggia, Massimo, Scortichini, Matteo, Hashizume, Masahiro, Honda, Yasushi, Hurtado-Díaz, Magali, Cruz, Julio, Nunes, Baltazar, Teixeira, João P., Kim, Ho, Tobias, Aurelio, Íñiguez, Carmen, Forsberg, Bertil, Åström, Christofer, Ragettli, Martina S., Guo, Yue-Leon, Chen, Bing-Yu, Bell, Michelle L., Wright, Caradee Y., Scovronick, Noah, Garland, Rebecca M., Milojevic, Ai, Kyselý, Jan, Urban, Aleš, Orru, Hans, Indermitte, Ene, Jaakkola, Jouni J.K., Rytí, Niilo R.I., Katsouyanni, Klea, Analitis, Antonis, Zanobetti, Antonella, Schwartz, Joel, Chen, Jianmin, Wu, Tangchun, Cohen, Aaron, Gasparrini, Antonio, & Kan, Haidong. Ambient Particulate Air Pollution and Daily Mortality in 652 Cities. *N. Engl. J. Med.* 381, 705–715 (2019).

Burnett, Richard, Chen, Hong, Szyszkowicz, Mięczyslaw, Fann, Neal, Hubbell, Bryan, Pope, C. Arden, Apte, Joshua S., Brauer, Michael, Cohen, Aaron, Weichenthal, Scott, Coggins, Jay, Di, Qian, Brunekreef, Bert, Frostad, Joseph, Lim, Stephen S., Kan, Haidong, Walker, Katherine D., Thurston, George D., Hayes, Richard B., Lim, Chris C., Turner, Michelle C., Jerrett, Michael, Krewski, Daniel, Gapstur, Susan M., Diver, W. Ryan, Ostro, Bart, Goldberg, Debbie, Crouse, Daniel L., Martin, Randall V., Peters, Paul, Pinault, Lauren, Tjepkema, Michael, Van Donkelaar, Aaron, Villeneuve, Paul J., Miller, Anthony B., Yin, Peng, Zhou, Maigeng, Wang, Lijun, Janssen, Nicole A.H., Marra, Marten, Atkinson, Richard W., Tsang, Hilda, Thach, Thuan Quoc, Cannon, John B., Allen, Ryan T., Hart, Jaime E., Laden, Francine, Cesaroni, Giulia, Forastiere, Francesco, Weinmayr, Gudrun, Jaensch, Andrea, Nagel, Gabriele, Concin, Hans, & Spadaro, Joseph V. Global estimates of mortality associated with longterm exposure to outdoor fine particulate matter. *Proc. Natl. Acad. Sci. U. S. A.* 115, 9592–9597 (2018).

b. In the results the authors state “Racial and ethnic disparities [in ZEV ownership and eVMT] are observed in both the DAVs and non-DACs, suggesting that a more targeted ZEV policy should be developed for BIPOC populations to overcome potential barriers such as linguistic isolation.” Why should equity in ZEV ownership be the policy target? If the goal is to reduce emissions or make greater emissions reductions in DACs that have worse air quality, shouldn’t the policy maker goal be to electrify the households with the highest VMT or emissions contributions within the DACs, regardless of origin? From an efficiency perspective, wouldn’t it make sense to think about minimizing the policy cost per ZEV adoption, or minimizing the policy cost per unit of pollution reduction resulting from ZEV adoption? If the goal is financial equity, aren’t there more efficient and cost-effective redistribution policies than subsidizing ZEVs?

Thank you for your thoughtful comments. From an emissions reduction standpoint, it’s certainly logical to focus electrification efforts on households contributing most to emissions. However, we also want to point out the importance of a just transition to clean transportation. In this broader context, efforts should be made to promote ZEV ownership across all socio-economic levels, ensuring equitable transition and leaving no

one behind. As studies have demonstrated, communities of color and residents of disadvantaged areas have historically endured disproportionate levels of traffic pollution. Redirecting more rebates and incentives towards these communities is a step towards achieving ZEV distributive justice and ensuring a just transition to clean transportation. We have expanded upon this in the last paragraph in the Discussion section, providing additional context and revising our recommendations accordingly. Revised sentences are provided below:

“To reduce this disparity, it is critical to ensure a just transition to clean transportation^{67,68}. As shown in our 2035 simulation results, the disparity can be reduced with more light-duty ZEVs. Although a universal ZEV incentive program can boost ZEV adoption and benefit DACs, targeted policies are needed to reduce the TRAP exposure gap between DACs and non-DACs, a result of historically unjust land-use policies. Recognizing and rectifying these historical injustices is a cornerstone of a just transition. This can be achieved by directing more rebates and incentives towards disadvantaged communities, providing them with opportunities to access clean transportation. To continue narrowing the gap, future policies and incentive programs should not only focus on DAC residents, but also tackle non-tailpipe emissions and specifically target trucks. These vehicles emit higher levels of pollutants and often travel through DACs^{58,69,70}. By adopting this comprehensive approach, we are taking a decisive step towards ZEV distributive justice and ensuring a just transition to clean transportation.”

c. Do the results support the existing ZEV incentive structure in California (e.g., substantially higher incentives for lower income households)?

Thank you for your comment. Whether our results support California's current ZEV incentive structure is beyond the scope of the current study. This is because the current manuscript mainly focused on near-roadway air quality resulting from ZEV adoption. To fully assess the effectiveness of ZEV incentive structure in California, a more comprehensive analysis is needed. This analysis should consider additional aspects associated with ZEV adoption, such as the just transition to clean transportation, as mentioned in our previous responses. Those topics are beyond the scope of our current paper which certainly warrants future study. To address this important comment and broaden the context of discussion, we have cited additional papers that discuss this issue in the revised manuscript. One such paper (Hennessy, 2023) finds that disparities in rebate distribution surpass those found in electric vehicle adoption in California.

Reference:

Hennessy, Eleanor M., & Syal, Sita M. Assessing justice in California's transition to electric vehicles. *iScience* 26, 106856 (2023).

6. The authors assume in their analysis that ZEVs are a perfect substitute for ICEVs (though they do not explicitly state, explain, or justify this assumption, which they should). Again, the authors have failed to examine or cite the relevant literature. Numerous studies have shown that when a household purchases a ZEV, they drive it differently. The results from the simulated air pollution model are therefore likely significantly biased.

a. Most ZEV households own more than one car and can substitute across vehicles. See, for example, Davis (2021).

b. ZEVs tend to be driven fewer miles than ICEVs. See, for example Burlig et al. (2021). Also, the 2017 National Household Travel Survey data show that on average across the US, ICEVs are driven 10,790 miles per year, and BEVs are driven only 7,040 miles per year.

Thank you for your comment. We acknowledge that as of 2020, households owning ZEVs and the drivers of these vehicles exhibit certain unique characteristics, which might evolve in the future. We analyzed drivers with daily travel distances over 200 miles, assuming that ZEVs would less likely be used for these extensive commutes. Our findings reveal that a mere 4% of travelers exceed this daily distance, suggesting that applying a distance filter for EV drivers would not result in a significant difference.

However, our ability to precisely model ZEV-specific trips is hindered by the current lack of quantitative data, despite our understanding of the different driving patterns of ZEVs. In fact, we are not aware of any practical dataset specifically for ZEV trips. We agree that our model might not fully capture the nuances of ZEV usage, and we have included the references you recommended and addressed these limitations in the Limitation subsection.

“The prediction of ZEV trips requires additional survey data encompassing household-level choices and distinct driving patterns^{90,91}. This includes preferences related to ZEV purchase, the availability of private and public EV charging stations, information on the distribution of ZEV incentives, and actual ZEV trip data that could be used to validate our model. Future research is warranted to address these aspects to improve the accuracy of ZEV-trip simulations. “

7. In their analysis, the authors use annual daily average concentrations of air pollutants, which I don't think is the best way to capture health impacts. My understanding of the literature that it's exposure that matters more- e.g., having a high dose for half the time and a low dose for half the time is worse than having a medium dose the whole time. Indeed, the EPA's NAAQS are crafted to be inline with epidemiological evidence on pollution exposure. I don't think the authors' approach captures this. I'd like to see them use an alternative measure- perhaps fraction of time with air pollution above a certain level or in non-compliance with NAAQS.

Thank you for your comment, which reflects the multifaceted nature of air pollution related health effects.

Air pollution is associated with both short-term and long-term health effects. This dual nature is reflected in the U.S. Environmental Protection Agency's (EPA) National Ambient Air Quality Standards (NAAQS), which stipulate both short-term and long-term exposure limits for various pollutants. For example, the NAAQS has primary and secondary standards for PM_{2.5}. The annual average standards are at levels of 12.0 µg/m³ and 15.0 µg/m³ respectively, while the 24-hour standards are based on the 98th percentile at 35 µg/m³ (<https://www.epa.gov/pm-pollution/national-ambient-air-quality-standards-naaqs-pm#rule-summary>). For NO_x, the NAAQS has set a 1-hour standard at a level of 100 ppb, based on the 3-year average of the 98th percentile of the yearly distribution of 1-hour daily maximum concentrations, as well as an annual standard set at a level of 53 ppb (<https://www.epa.gov/no2-pollution/primary-national-ambient-air-quality-standards-naaqs-nitrogen-dioxide#rule-summary>).

The dose-response function, or the relationship between the level of exposure to a pollutant and the severity of health outcomes, is complex (see figures under response No. 5). Even exposure to low levels of pollution can be harmful as we mentioned above. Some studies have suggested that sustained exposure to lower levels of certain pollutants can lead to serious health consequences over the long term, such as an increased risk of cancer (Hvidtfeldt, 2021). Conversely, high levels of exposure over a short period could induce immediate adverse health effects, such as an asthma attack, leading to increased asthma-related mortality (Liu, 2019).

Your suggestion to examine the fraction of time that pollution levels exceed a certain threshold corresponds with EPA's short-term standards, which focus on high exposures over brief periods. However, our study primarily focuses on the long-term health effects of air pollution exposure. This is particularly relevant for assessing the experiences of individuals living in DACs, who are likely exposed to air pollution in their residential environments over extended periods of time. In this context, the use of annual average daily concentrations is a scientifically sound and commonly accepted methodology in air pollution research. This approach aligns with the EPA's long-term exposure standards.

While we appreciate and acknowledge the importance of the suggestion, it is beyond the scope of the current work. We agree that examining different exposure metrics, including short-term high exposure, would yield valuable insights and suggest this as a direction for future research. We have added these points in the Limitation sub-section.

“Third, while the utilization of annual average daily concentrations is a scientifically sound approach commonly employed in air pollution research and aligns with EPA’s long-term exposure standards, we acknowledge that air pollution is associated with both short-term and long-term health effects. Future studies could enhance our understanding by examining these effects at a more granular temporal scale, leveraging our transportation model, which has temporal resolution down to the hour.”

References:

Hvidtfeldt, Ulla Arthur, Severi, Gianluca, Andersen, Zorana Jovanovic, Atkinson, Richard, Bauwelinck, Mariska, Bellander, Tom, Boutron-Ruault, Marie Christine, Brandt, Jørgen, Brunekreef, Bert, Cesaroni, Giulia, Chen, Jie, Concin, Hans, Forastiere, Francesco, van Gils, Carla H., Gulliver, John, Hertel, Ole, Hoek, Gerard, Hoffmann, Barbara, de Hoogh, Kees, Janssen, Nicole, Jöckel, Karl Heinz, Jørgensen, Jeanette Therning, Katsouyanni, Klea, Ketznel, Matthias, Klompmaker, Jochem O., Krog, Norun Hjertager, Lang, Alois, Leander, Karin, Liu, Shuo, Ljungman, Petter L.S., Magnusson, Patrik K.E., Mehta, Amar Jayant, Nagel, Gabriele, Oftedal, Bente, Pershagen, Göran, Peter, Raphael Simon, Peters, Annette, Renzi, Matteo, Rizzuto, Debora, Rodopoulou, Sophia, Samoli, Evangelia, Schwarze, Per Everhard, Sigsgaard, Torben, Simonsen, Mette Kildevæld, Stafoggia, Massimo, Strak, Maciek, Vienneau, Danielle, Weinmayr, Gudrun, Wolf, Kathrin, Raaschou-Nielsen, Ole, & Fecht, Daniela. Long-term low-level ambient air pollution exposure and risk of lung cancer – A pooled analysis of 7 European cohorts. *Environ. Int.* 146, (2021).

Liu, Yuewei, Pan, Jingju, Zhang, Hai, Shi, Chunxiang, Li, Guo, Peng, Zhe, Ma, Jixuan, Zhou, Yun, & Zhang, Lan. Short-term exposure to ambient air pollution and asthma mortality. *Am. J. Respir. Crit. Care Med.* 200, 24–32 (2019).

8. Though the authors briefly mention upstream emissions in the limitations section, this is too important an issue to hand-wave away. The authors claim that changes in upstream emissions are likely to be negligible due to the low ZEV penetration rate, however, I don't see why the same argument wouldn't apply to their main analysis. Furthermore, to the extent the implications of the paper are relevant for future policy recommendations, changes in upstream emissions are only going to become more important in the future as ZEV penetration rates increase. Also, there is usually an environmental justice issue associated with electricity generation, with more disadvantaged populations more likely to be located near power plants. It would be very interesting for the authors to incorporate upstream emissions into their analysis. At a minimum, they should perform some back of the envelope calculations or cite some related literature here to assess how they might impact their findings.

Thank you for raising the important issue of upstream emissions. In the field of air quality research, there is a distinction between regional and near-roadways air pollutants. Our focus in this study is on the near-roadway, traffic-emitted pollutants. Upstream emissions, in contrast, often originate from point sources, which produce high plume rises and contribute to regional pollutants across a large geographic area. When we state “negligible”, we mean that upstream emissions have minimal impact on traffic-emitted air pollutant concentrations near-roadways. We have revised our expression in the Limitation section.

Regarding future scenarios, we don't expect upstream emissions would significantly affect near-roadway pollution either, or even regional air pollutant concentration. In one of our previous studies, we assessed the repowering of several power plants in the Los Angeles area, anticipating increased electricity needs by 2030. We found that the contributions from upstream emissions due to additional electricity demand contribute

less than 1% of SO₂, NO_x, and PM_{2.5}, suggesting they exert minimal effects on regional air quality in the Los Angeles Basin. This can be attributed to stricter emission standards in the future, particularly in Los Angeles and California. Current plans for electricity generation units involve phasing out fossil fuels in favor of clean energy. Moreover, for the remaining power plants that help balance the power grid load, the city is planning to transition to hydrogen instead of fossil fuels.

In addition, the LA100 study conducted by the National Renewable Energy Laboratory (NREL) supports our findings. Their future projection in 2045, which accounted for even more electricity demands from ZEVs, found “significant reductions in exposure to air pollutant emissions from the LADWP facilities” for disadvantaged populations living near power plants (Hettinger et al., 2021, p. 67). This prediction is again due to stricter regulations for future upstream emissions, as mentioned earlier. As we have mentioned in our manuscript, we believe that a clean energy portfolio, which is the case in Los Angeles and California, is the key to reducing upstream emissions and ensuring environmental justice.

We added following sentences and associated references that support such a claim in the revised manuscript.

“Finally, we acknowledge the impacts of upstream emissions from electricity generating units resulting from increased electricity demand due to ZEVs. However, we anticipate that these emissions would have minimal impact on TRAP near roadways. Nevertheless, studies have suggested that a clean energy portfolio is the key to reducing upstream emissions and ensuring environmental justice^{3,57}. Reducing emissions from electricity generating units could yield air quality benefits, even with the increased energy demand from ZEVs^{93,94}. Shifting from fossil fuels to clean energy sources has the potential to substantially reduce the exposure of disadvantaged populations to air pollutants emitted from these units, particularly those living in close proximity⁹⁵.”

Reference:

Hettinger, Dylan, Jaquelin Cochran, Vikram Ravi, Emma Tome, Meghan Mooney, and Garvin Heath. 2021. “Chapter 10: Environmental Justice.” In *The Los Angeles 100% Renewable Energy Study*, edited by Jaquelin Cochran and Paul Denholm. Golden, CO: National Renewable Energy Laboratory. NREL/TP-6A20-79444-10. <https://www.nrel.gov/docs/fy21osti/79444-10.pdf>.

Reviewer #3:

Synopsis: The authors study the distribution of ZEVs in CA/LA county, simulate their link level traffic patterns, and estimate changes in both emissions and pollutant concentrations near roadways. They find that ZEVs have benefited everyone and that the benefits in DACs have been greater, but not so great as to erase systemic and historical disadvantages.

Recommendation: This is a mostly well-written and well-presented study, however, given that it is based on historical changes of just ~2% ZEV adoption, I would contend that the paper would be immensely more impactful if the authors added forward looking simulations of higher adoption percentages. In particular, since the authors have chosen a framework that presents ZEVs as failing to offset longstanding injustices, it seems ideal/critical to include simulations that demonstrate the magnitude and nature of adoption rates required to level the exposure playing field.

Comments and Criticisms:

- The work presented here is immensely relevant. In fact, something similar was published recently: <https://doi.org/10.1016/j.scitotenv.2023.161761> that should be cited here and incorporated into the discussion.

We appreciate you sharing this recent paper with us. We have incorporated it into our Discussion section as follows:

“Previously, a study⁶⁶ reported an association between zip code level ZEV adoption and lower ambient nitrogen dioxide concentrations albeit not statistically significant. By tracking individual trip and utilizing link-level emission data specifically from traffic, our innovative approaches allow us to model the near-roadway air quality benefits attributable to ZEVs, finding statistically significant differences.”

- A second recent study using a much more sophisticated CTM, is cited by the authors, i.e., Skipper et al. (2023). However, its use in the current manuscript is quite limited and curious. For example, the study is only referenced with regard to PM changes, and it is stated that Skipper et al’s 100% ZEV simulation results “match” the 2.2% simulation PM magnitude changes presented here. Can the authors explain how these changes match? Given the greater sophistication of Skipper et al.’s CTM simulations, it would be helpful if the authors were to discuss what is and is not achievable with their modeling set up verses that of Skipper et al. For example, Skipper et al. point out that ZEV adoption increases ozone concentrations, a finding previously reported by Pan et al. (2019) in Houston and Peters et al. (2020) in some CONUS locales (both studies are cited by the authors).

Thank you for your comments. In our original manuscript, the term "match" was used to denote a similar trend rather than a direct numerical match. With the additional 2035 scenarios we simulated in the revised manuscript, we are now able to compare our

results more directly with the results from Skipper et al. We have now revised the second last paragraph in the Results section as follows:

“A recent study³³ simulating full electrification of light-duty vehicles and buses in California has demonstrated an average PM_{2.5} reduction of 0.13 µg/m³. These findings corroborate our results for a 100% ZEV scenario for light-duty vehicles in 2035, which results in an PM_{2.5} reduction of 0.10 µg/m³ for DAC (see Table S1).”

Regarding ozone, we understand that ZEV adoption, which reduces NO_x emissions, could increase ozone concentrations, especially in VOC-limited regions such as Los Angeles County. This aligns with the findings of the LA100 report conducted by NREL, which we have now included in our revised manuscript in the Discussion section. However, since our study primarily focuses on near-roadway air quality and considering the presence of NO, which readily reacts with ozone to form NO₂, ozone pollution is not a concern in close proximity to roadways. Instead, ozone, as a secondary air pollutant, is only formed through atmospheric chemical reactions between VOC and NO_x much further downwind from roadways. This is why the EPA's near-road monitoring stations do not track ozone as its concentration is typically very low near-roadways (<https://www.epa.gov/amtic/near-road-monitoring>).

However, we understand the significance of highlighting the potential impact of ZEV adoption on ozone at the regional level. As such, we have incorporated a discussion on this matter in the third paragraph in the Discussion section. This additional section aims to provide a more holistic view of ZEV adoption's potential consequences.

“While this decline in disparity is promising near roadways, regional air quality and secondary pollutants such as ozone will require more attention in the future. Owing to the complex nature of ozone⁵⁹, ZEV adoption, which reduces NO_x emissions, could paradoxically increase ozone concentrations, especially in VOC-limited regions such as Los Angeles County. This has been reported by the LA100 study⁶⁰ conducted by the National Renewable Energy Laboratory. Future strategies will need to consider ozone concentrations in a region-specific context.”

- The authors have chosen to frame the standard by which to judge EVs, as equal exposure to pollutants for all populations. I think a reasonable person would agree. Given this framing, and the conclusion that EVs fail this threshold, the manuscript would be immensely more impactful if the authors were able to demonstrate that this threshold could be attained with EVs alone. Perhaps it cannot, in which case maybe that's not the best metric of success against which to judge EVs? However, if there is a scenario of incentives and fractional adoption that results in exposure equity, these authors seem well placed to identify it. Demonstrating this solution would have much more tangible policy relevance than the current results.

We appreciate your insightful suggestion. In response, we have further explored two potential scenarios for 2035: one in which 50% of light-duty vehicles are ZEVs based on the Mobile Source Strategy report published by California Air Resource Board, and

another extreme scenario where 100% of light-duty vehicles are ZEVs. Both scenarios also incorporate a 16% medium-duty ZEV presence and a 20% heavy-duty ZEV presence, as projected by the EMFAC database. Our updated findings as shown in new Table 1, Table S1, and Figures 5-6 indicate that as the proportion of light-duty ZEVs increases, the disparity in NO_x pollution between DACs and non-DACs reduces. PM_{2.5} emissions also decrease, albeit at a lesser rate due to the contribution of brake and tire wear emissions. This shows that while ZEVs are part of the solution, they alone cannot fully resolve the disparity. As a result, our study highlights the need for a broader range of strategies, including the electrification of trucks and a focus on non-tailpipe emissions. We have integrated these additional insights into the last paragraph in the Discussion section as follows.

“To continue narrowing the gap, future policies and incentive programs should not only focus on DAC residents, but also tackle non-tailpipe emissions and specifically target trucks that emit higher levels of pollutants and travel through DACs.”

- The omission of a conversation about ozone is likely driven by their air quality model limitations, but since the subject of the manuscript is TRAPs (i.e., traffic-related air pollution) discussion of ozone seems like a relevant discussion point. In particular, the authors posit on Lines 317-320 that their model design is easily generalizable to other locales, but perhaps this is not so true when one wants to assess the holistic air quality impacts of ZEVs, as in the case of ozone, given CONUS-wide differences in VOC to NO_x regimes.

We agree with your comment regarding the importance of ozone. However, ozone is not typically considered a near-roadway pollutant due to its secondary formation nature as we explained above. Nevertheless, we think it is necessary to mention ozone in the context of ZEV adoption. As such, we have incorporated a discussion on this matter as shown in the previous response.

As for the generalizability of our model design, we appreciate your insight. Given the historical context of urban planning and racial segregation from the 1960s to the 1990s in the entire CONUS, our near-roadway model approach and findings on excessive near-roadway PM_{2.5} and NO_x exposure to BIPOC can be generalized to other metropolitan areas.

We agree that to study ozone pollution, researchers need to consider regional variations in VOC to NO_x regimes. For instance, the LA100 study conducted by NREL and our own Chemical Transport Model from another project indicate that in VOC-limited regimes like Los Angeles, reducing NO_x will increase ozone concentrations. We have thus included a discussion on ozone pollution in our manuscript, citing related literature to suggest that ozone concentrations might need to be considered based on regional features, as shown in the previous response.

- Have the authors considered population-weighting their concentration changes?

Thank you for your suggestion. In our study, we have aggregated the near-roadway pollutant concentrations to the census tract level in order to analyze the differential impacts on DACs and non-DACs. Census tracts have similar population levels typically around 4,300 residents according to the Census Bureau. We also found no statistically difference between populations in DAC and non-DAC (see data below). Thus, we did not perform population-weighted concentration changes, as other studies that utilize grid-based geographic units often do.

Data:

All: N=2341, population: 4305 ± 1579

DAC: N=1173, population: 4318 ± 1481

non-DAC: N= 1168, population: 4291 ± 1672

- I did not find validation information for the baseline R-Line simulations.

Thank you for your valuable comments. The R-Line model is a well-established dispersion modeling system used extensively for near-road air quality analysis. It was developed by the U.S. EPA and has received thorough verification and validation. This model has been widely tested against observed data in a range of environments and proven reliable in numerous studies (Wu et al., 2011; Milando et al., 2018). The historical validation and certification from the EPA give us confidence in the accuracy of the outcomes it provides.

Our data in the manuscript focus on traffic-emitted pollutants in Disadvantaged Communities (DAC) and non-DAC, without considering the background pollutant concentration. It is thus difficult to use ambient monitoring data which include background levels to directly validate our simulation results. To bridge this gap, we found a recent study that analyzed the relationship between reduced traffic flow and the consequential decline in near-roadway NO and NO₂ concentrations in California during the COVID-19 pandemic (Liu et al., 2021). This COVID-19 study utilized data from EPA's near-roadway monitoring stations including the monitoring station located in Long Beach Route 710 Near Road which is within our modeling domain. As per the findings of Liu et al., (2021), a $22 \pm 12\%$ decrease in passenger traffic and a $7 \pm 6.1\%$ reduction in truck traffic on weekdays correlated to a significant drop in NO and NO₂ concentrations, by $47 \pm 8.5\%$ and $15 \pm 6.4\%$, respectively (Table 1 and Figure 2 of Liu et al., (2021)), which translates to a ~24-38% reduction of NO_x when taking into account both NO and NO₂ reductions. To validate R-Line simulation, we ran scenarios simulating normal traffic flow, followed by the same traffic reductions as reported in Liu et al., (2021). Our results show a $25 \pm 14\%$ reduction in NO_x emissions which agrees well with the referenced study's findings, providing good validation for the R-Line simulations.

References:

Wu, J., Wilhelm, M., Chung, J. & Ritz, B. Comparing exposure assessment methods for traffic-related air pollution in an adverse pregnancy outcome study. *Environ. Res.* 111, 685–692 (2011).

Milando, C. W. & Batterman, S. A. Operational evaluation of the RLINE dispersion model for studies of traffic-related air pollutants. *Atmos. Environ.* 182, 213–224 (2018).

Liu, Jonathan, Lipsitt, Jonah, Jerrett, Michael, & Zhu, Yifang. Decreases in Near-Road NO and NO₂ Concentrations during the COVID-19 Pandemic in California. *Environ. Sci. Technol. Lett.* 8, 161–167 (2021).

- The authors summarize their emissions considered as: “running exhaust emissions, start exhaust tailpipe emissions, and brake and tire wear emissions” on Lines 418-419. Does this include idling emissions? And is the behavior of vehicles and its effect of emission rates considered in the emissions modeling? For example, the emission profile of a vehicle changes as it sits idling and cools.

Thank you for your comment. Idling emissions are included in our analysis based on data from the EMFAC (EMission FACtors) model, a tool developed by the California Air Resources Board (CARB). According to the EMFAC, idling emissions for light-duty vehicles are reported as zero for both NO_x and PM_{2.5}. For medium-duty vehicles and heavy-duty vehicles, we included idling emissions as reported by the EMFAC in units of grams per vehicle per day. In terms of emission profile and behavior, these factors have been taken into consideration within the EMFAC model, thus reflected in our analysis.

- In the discussion of R-Line, simulation of NO_x is mentioned, but the paper focuses on NO₂. Just to clarify, does R-Line simulate NO₂, NO_x, or both, and which is reported here?

In our study, the emission rates for NO_x, which we used in our simulations, were sourced directly from the EMFAC (EMFAC does not provide emission rates for NO₂ and NO separately). These NO_x emissions were then introduced into the R-Line model for our dispersion simulations. Therefore, the R-Line model in our study is specifically simulating the dispersion of NO_x, and all the values presented and discussed in our manuscript correspond to NO_x.

- On Line 375 it is said: “A truck was equivalent to 3.5 passenger cars.” What does this mean?

We adopted the concept of a "Passenger Car Equivalent" (PCE) to account for the greater disruption caused by longer and heavier trucks to traffic flow compared to conventional passenger cars. A PCE value of 3.5 was used, signifying that a truck has the same impact on traffic flow as 3.5 passenger cars. We have provided additional clarifications in the Methodology section:

“In the simulation, we used a "Passenger Car Equivalent" of 3.5, indicating that a truck impacts traffic flow equivalently to 3.5 conventional cars⁷⁷.”

Reference:

FHWA, (2000). Comprehensive Truck Size and Weight Study, Vol III Scenario Analysis, Chapter 9. Retrieved from: <https://www.fhwa.dot.gov/reports/tswstudy/Vol3-Chapter9.pdf>.

- On Lines 485-492, the topic of upstream emissions is broached. There are two studies that can support this text: <https://doi.org/10.1016/j.atmosenv.2019.04.003> & <https://doi.org/10.1021/acs.est.6b04868> although it may be important to point out that neither is of a sufficient resolution to assess equitable outcomes, as done in citation 66, albeit with an econometric model.

Thank you for providing these valuable references. We have reviewed both studies and included them in our discussion on upstream emissions, in the Limitation section as below.

“Finally, we acknowledge the impacts of upstream emissions from electricity generating units resulting from increased electricity demand due to ZEVs. However, we anticipate that these emissions would have minimal impact on TRAP near roadways. Nevertheless, studies have suggested that a clean energy portfolio is the key to reducing upstream emissions and ensuring environmental justice^{3,57}. Reducing emissions from electricity generating units could yield air quality benefits, even with the increased energy demand from ZEVs^{93,94}. Shifting from fossil fuels to clean energy sources has the potential to substantially reduce the exposure of disadvantaged populations to air pollutants emitted from these units, particularly those living in close proximity⁹⁵.”

REVIEWER COMMENTS

Reviewer #2 (Remarks to the Author):

The authors have generally done a good job of addressing the referees comments. I have several more comments related to their responses/edits and new additions:

[Regarding Comment 1.ii] Given that Holland et al. (2019) performs a similar analysis using a top-down approach, and a major contribution of this paper is doing a bottom-up approach, 1) how do the results compare and 2) what do we learn from this paper that we couldn't learn from Holland et al. (2019)?

[Regarding Comment 6] Given that empirical evidence/data exists as to how ZEVs are currently being driven, it is not sufficient to simply say that additional data are required. I would really like to see a robustness check of how the main results change assuming ZEV VMT is $\sim 2/3$ of ICEVs, since that is what current data points to. At the very least, the authors should include a couple of sentences on how they would expect this empirical fact to impact their results.

I like the addition of a scenario through 2035. However, the assumptions/model behind the scenario are too simplistic. While a logistic growth model may be appropriate on average, it is highly unlikely that the growth rate path will be similar across all census tracts, especially given potentially interesting dynamics in the used vehicle market and changing rates of dirty vehicle retirements (i.e., as the 2035 100% ZEV becomes binding, people will likely hold off on purchasing new vehicles, keeping dirty vehicles longer, and bolstering the used car market). (On a more minor note, the authors should clarify that the 2035 mandate applies only to new vehicles.) I expect a lot of heterogeneity across neighborhoods. Also, I cannot find the source of the 50% ZEVs in 2035 from reference 47 (CARB website). I do see a slide deck (https://ww2.arb.ca.gov/sites/default/files/2021-05/2020_MSS_May_Webinar_Presentation.pdf) showing a projection of 48% ZEV in the heavy-duty sector by 2037, but that's not passenger vehicles. Lastly, the same CARB projections assume decreasing VMT by 2035, which should also be factored into the authors' projections (with heterogeneity across census tracts?).

Reviewer #3 (Remarks to the Author):

I very much like this study and appreciate the authors responses to my queries. I particularly appreciate the bottom up approach and the addition of 2035 estimates. However, I have a few remaining questions/suggestions for the author's considerations.

The disparities paragraph on Lines 49-74 is quite good, as are subsequent results/discussion related to this topic, however I have a clarifying question centered on baseline exposure v. susceptibility: do the authors know which is the main driver of air quality impacts in their domains of interest. There is quite of bit of language dancing around this key consideration, but from a policy solution perspective it would be good to indicate if there is more value in programs that reduce pollutant exposure or in programs that decrease population susceptibility. I realize both contribute, but the highest levels of pollutants do not always coincide with BIPOC and low-income populations. Perhaps a key distinction here is that this analysis only deals with near-roadway pollution. If that is the case, a philosophical question: what is the appropriate spatial scale at which to assess population disparities; near-road, census tract, city, county, or state? It seems the choice of domain could substantially influence the equity determination. This is also a potential limitation of working with a dispersion model, as opposed to a regional CTM. Some added discussion to this last point would be helpful, as would an analysis to determine the initial question asked in this paragraph.

Line 307: given battery weights and potential increases in brake and tire wear, do your PM estimates

include greater PM emissions from these sources, or do you assume they are the same as the vehicles they replace? There's been conflicting findings in the literature, so either method is justified, but the assumption should be noted.

Typo on Line 343 "re".

For the ozone discussion on Lines 351-358, please refer to and cite Skipper et al (see their Figure 6).

Line 407-408: The authors may not have this data, but a question: from an air quality perspective, is it more effective to incentivize ZEV adoption in DACs or to incentivize reducing emission from or electrifying trucks? This question has real-world relevance to the IRA that prioritized passenger vehicle subsidies over reducing truck emissions.

Response to Reviewers' Comments

We appreciate the additional comments provided by the reviewers. All comments from the reviewers have been addressed below, and changes are highlighted in the revised manuscript.

Reviewer #2 (Remarks to the Author):

The authors have generally done a good job of addressing the referees comments. I have several more comments related to their responses/edits and new additions: [Regarding Comment 1.ii] Given that Holland et al. (2019) performs a similar analysis using a top-down approach, and a major contribution of this paper is doing a bottom-up approach, 1) how do the results compare and 2) what do we learn from this paper that we couldn't learn from Holland et al. (2019)?

Thank you for the question. 1) We have created a table below to compare the approach, data sources, assumptions, as well as results between Holland et al., (2019) and our study.

	Holland et al. (2019)	Our study
Study Approach	Top-down, assuming county air quality benefits distributed equally to each census block group	Bottom-up, using link-level traffic data for near-roadway air quality calculation for census block group
EV/ZEV Data Usage	Assume EVs only drive in the county where they are registered/purchased	Incorporate ZEV census tract registration data with trip routes; track ZEV trips across different tracts
VMT Assumption	Assumes both gasoline and EV drive 15,000 miles/year	Uses VMT based on simulated person-level trips; assume equal VMT for ICEV and ZEV
Scenario	A base year scenario	A base year scenario and two 2035 policy scenarios
Vehicle Fleet	Passenger vehicle	Passenger vehicle and truck
Air Quality Focus	Regional county level	Near-roadway
Air Quality Model	AP2 (Regional reduced-complexity model)	R-LINE (Near roadway dispersion model)
Environmental Impact	Positive impacts for EV in CA but negative in other US regions	Positive near roadway impacts in CA regardless of ZEV registration tracks
Equity Findings	Positive environmental benefits for block groups with median income > \$65,000; negative for those below this threshold	Disadvantaged community census tracts with fewer ZEVs still receive high absolute air quality benefits, albeit with low relative reduction
Ethnoracial Findings	Positive benefits for Hispanic residents; negative for white residents	Whites receive a higher-than-average emission reduction share, whereas Hispanics receive less than their population share

2) As shown in the table above, Holland et al. (2019) employed a top-down approach, and assumed that damages from gasoline vehicles and benefits from electric vehicles

were uniformly distributed across all census block groups within a county. This top-down approach ties ZEV-related air quality benefits to where ZEVs are registered, and doesn't account for variations of highway density, socio-economic characteristics, and the frequency of ICEV/ZEV trips across different census tracts.

In contrast, our bottom-up approach is based on individual ZEV trips which ties ZEV-related air quality benefits to where ZEVs are driven. This distinction in methodology provides an explanation for the observed \$65,000 income threshold in Holland et al.'s findings. They observed that census block groups with median incomes above this threshold enjoyed positive environmental benefits, while those below faced adverse effects. Yet, our study reveals that even disadvantaged communities can benefit from ZEVs registered elsewhere, as they traverse various tracts and improve near-roadway air quality. Additionally, we incorporated ZEV trucks and forecasted 2035 scenarios, enriching our policy implications which were not covered in Holland et al.'s paper.

We have updated the second paragraph in our Discussion section as follows:

“...This finding is encouraging in that ZEVs can offer near-roadway air quality benefits to various communities. Unlike previous research⁵⁸ that focused on regional air quality benefits from ZEVs using a top-down approach—based on vehicle registration locations—our study employs a bottom-up methodology centered on actual ZEV trip routes. Consequently, we discovered that near-roadway air quality benefits can be distributed irrespective of DAC designation, a novel insight not addressed in the literature.”

[Regarding Comment 6] Given that empirical evidence/data exists as to how ZEVs are currently being driven, it is not sufficient to simply say that additional data are required. I would really like to see a robustness check of how the main results change assuming ZEV VMT is ~2/3 of ICEVs, since that is what current data points to. At the very least, the authors should include a couple of sentences on how they would expect this empirical fact to impact their results.

We appreciate your comment and dive deeper into ZEV VMT data. We found while empirical data do exist, they are not always consistent. For example, in Burlig et al. (2021), the authors indicated that Tal et al. (2020) reported completely different survey results than the 2017 NHTS results. The study by Tal et al. (2020) states that annual VMT estimates from their nationwide survey show that BEV owners drive on average more than 10,000 miles annually. PHEV mileage could be even higher than ICEV based on some survey results (see the table below for comparison of the annual VMT from different surveys:

[redacted]

(Figure adopted from Tal et al. 2020, page 56-57)

While the eVMT will affect the absolute pollution reduction value, the main results of our study, which focus on equity, will not change. This is because any eVMT change would be universal for both DAC and non-DAC. It won't impact our equity analysis—the relative shares of eVMT and disparities between DAC and non-DAC would remain the same. To make it clear, we have now included our assumptions in the 'ZEV trip assignment and emission calculation' subsection under the 'Methods' section as follows:

“We calculate emissions in Los Angeles County for both 2020 and 2035, with and without ZEVs. For our 2020 estimate, we use real-world ZEV ownership data from CARB to determine traffic emissions. For 2035, we rely on the projected ZEV ownership data. We assume that both ICEVs and ZEVs have the same vehicle miles traveled (VMT), indicating similar driving behaviors and patterns for both vehicle types. To establish a baseline for our study, we consider all on-road light-duty vehicles to be ICEVs, thus excluding ZEVs.”

We have also added more discussion regarding this question into the 'Limitation' section as follows:

"Second, we assume that ZEVs and ICEVs have identical VMTs. Although we recognize potential variations in driving patterns between ZEV and ICEV drivers, current empirical data yield inconsistent conclusions, making it difficult to adjust our model. The average annual VMT for ICEV is between 11k and 12k miles, while for ZEVs, it ranges from 6k to 15k miles, depending on survey and modeling methods^{92, 95}. Moreover, most

existing empirical data are aggregated, typically at the annual level, which is inadequate to calibrate our agent-based simulation that requires detailed driving log data. Variations in eVMT might affect absolute pollution reduction values, but the relative shares of eVMT and disparities between DAC and non-DAC remain consistent. Thus, our equity-focused findings will not be affected.”

Reference:

Burlig, Fiona, Bushnell, James, Rapson, David, & Wolfram, Catherine. Low Energy: Estimating Electric Vehicle Electricity Use. *AEA Pap. Proc.* 111, 430–435 (2021).

Tal, Gil, Srinivasa Raghavan Vaishnavi Chaitanya Karanam Matthew Favetti Katrina May Sutton Jade Motayo Ogunmayin Jae Hyun Lee, Seshadri P, Nitta, Christopher, Kurani, Kenneth, Chakraborty, Debapriya, Nicholas, Michael, & Turrentine, Tom. Advanced Plug-in Electric Vehicle Travel and Charging Behavior Final Report (CARB Contract 12-319-Funding from CARB and CEC). (2020).

I like the addition of a scenario through 2035. However, the assumptions/model behind the scenario are too simplistic. While a logistic growth model may be appropriate on average, it is highly unlikely that the growth rate path will be similar across all census tracts, especially given potentially interesting dynamics in the used vehicle market and changing rates of dirty vehicle retirements (i.e., as the 2035 100% ZEV becomes binding, people will likely hold off on purchasing new vehicles, keeping dirty vehicles longer, and bolstering the used car market). (On a more minor note, the authors should clarify that the 2035 mandate applies only to new vehicles.) I expect a lot of heterogeneity across neighborhoods.

Thank you for your comment. We appreciate your insights on our 2035 scenario assumptions. Our logistic model is indeed specific to individual census tracts, using 2015 to 2020 ZEV data from each census tract to initiate the logistic growth model, as described in the first revision line 435. To clarify this point, we have now revised the manuscript as follows:

“To project the future ZEV adoption in 2035 when all new passenger vehicles sold are expected to be ZEVs, we applied different logistic growth models to estimate the number of light-duty ZEVs for each census tract within Los Angeles County based on census tract specific historical ZEV adoption data between 2015 to 2020.”

Forecasting the future is inherently complex. While there are factors that could negatively affect the used car market dynamics and the retirement of older and dirty vehicles, there are also policies or programs that could positively influence these dynamics. For instance, California could potentially expand its existing vehicle early retirement programs offering more incentives to reduce the number of older and dirty vehicles. We agree that heterogeneity would undoubtedly exist in each neighborhood, which is why we formulated different logistic growth models for individual census tracts. However, due to gaps in available data, such as detailed insights into the used car

market dynamics at the census tract level or data on how each tract reacts to policy incentives, it is beyond the scope of our study. Overall, we believe our census-specific logistic growth model is sufficient for the purpose of near-roadway air quality analysis.

Regarding the minor note. We have revised the manuscript to the following to make it clear that the 2035 mandate applies only to new vehicles:

“Executive Order N-79-20 of September 2020 requires all new passenger vehicles sold in California to be ZEVs by 2035⁷. In 2022, the CARB approved the Advanced Clean Cars II rule, which establishes a year-by-year roadmap so that by 2035 100% of new cars and light trucks sold in California will be ZEVs⁸.”

Also, I cannot find the source of the 50% ZEVs in 2035 from reference 47 (CARB website). I do see a slide deck (https://ww2.arb.ca.gov/sites/default/files/2021-05/2020_MSS_May_Webinar_Presentation.pdf) showing a projection of 48% ZEV in the heavy-duty sector by 2037, but that’s not passenger vehicles.

Regarding the specific reference from CARB, in our 2035 scenario, the light-duty vehicle ZEV figures are derived from the MSS website's Vision model, specifically from the "Update to the Board - December 10, 2020" section in the CARB MSS website. The source file, titled 'LDV_MSS_supporting_materials_ISAS_Nov2020.xlsx,' estimates a combined total of 15 million PHEVs, BEVs, and FCEVs in 2035 under the 'population' tab, from a total vehicle count of 30 million. This yields our 50% ZEV figure.

The medium- and heavy-duty ZEV numbers are obtained from the META Online Tool, applying the MSS default parameters as provided in the "Webinar - May 6, 2021" section. We appreciate your bringing this up, and we've now incorporated these references into our manuscript in addition to the MSS main webpage. See references below:

References:

California Air Resources Board. Vision Model - LDV Raw Data and Results. https://ww2.arb.ca.gov/sites/default/files/2020-11/LDV_MSS_supporting_materials_ISAS_Nov2020.xlsx (2020).

California Air Resources Board. META Online Tool. <https://arb.ca.gov/emfac/meta/> (2021).

Lastly, the same CARB projections assume decreasing VMT by 2035, which should also be factored into the authors’ projections (with heterogeneity across census tracts?).

We are aware of the CARB’s 2022 Scoping Plan, which sets a goal to reduce VMT per capita by 25% below 2019 levels by 2030, and 30% below 2019 levels by 2045 (CARB, 2022). Yet, within the same document, CARB highlights the inherent regulatory challenges associated with achieving this target and pointed out these targets are not regulatory requirements. Unlike mandates for new ZEV sales, imposing strict VMT

regulations is almost impossible. This complexity stems from historically established transportation and land use policies, which are deeply entrenched and difficult to modify. In fact, this isn't the first time CARB has aimed to reduce VMT. For instance, the 2017 CARB Scoping Plan also sought to reduce the VMT, yet the actual VMT trend continues to rise (<https://ww2.arb.ca.gov/resources/documents/carb-2017-scoping-plan-identified-vmt-reductions-and-relationship-state-climate>). As a result, we chose not to factor this VMT trajectory into our model, particularly given the historical trend of rising VMT despite intentions to curtail it (see VMT trend from CA DOT).

[redacted]

(Figure adopted from CA DOT <https://dot.ca.gov/programs/sustainability/sb-743>)

Reference:

CARB. (2022). 2022 Scoping Plan for Achieving Carbon Neutrality. <https://ww2.arb.ca.gov/sites/default/files/2023-04/2022-sp.pdf>

Reviewer #3 (Remarks to the Author):

I very much like this study and appreciate the authors responses to my queries. I particularly appreciate the bottom up approach and the addition of 2035 estimates. However, I have a few remaining questions/suggestions for the author's considerations.

The disparities paragraph on Lines 49-74 is quite good, as are subsequent results/discussion related to this topic, however I have a clarifying question centered on baseline exposure v. susceptibility: do the authors know which is the main driver of air

quality impacts in their domains of interest. There is quite of bit of language dancing around this key consideration, but from a policy solution perspective it would be good to indicate if there is more value in programs that reduce pollutant exposure or in programs that decrease population susceptibility. I realize both contribute, but the highest levels of pollutants do not always coincide with BIPOC and low-income populations. Perhaps a key distinction here is that this analysis only deals with near-roadway pollution.

Thank you for your comments which accurately highlight that both exposure and susceptibility are critical determinants of air quality health outcomes. Our analysis, as you noted, primarily focus on near-roadway pollution, representing the exposure aspect. Deciding to emphasize exposure over susceptibility or vice versa warrants a comprehensive cost-benefit analysis, which is beyond the scope of our current study. Performing a cost-benefit analysis for policies that might mitigate susceptibility can be complex. Factors to consider include diverse health outcomes, a range of pollutant types (e.g., air or water; and if air, then indoor vs. outdoor, near-road vs. regional), and a variety of potential policy interventions. These interventions could span from enhancing healthcare access, championing equitable housing, fostering healthy behaviors, investing in green spaces, bolstering community resilience, or bridging income and educational gaps.

Even within the air pollution domain, there is a lack of robust methodologies to quantify specific susceptibility. As a result, the literature remains limited. Among these limited studies, many focus broadly on the general population without delving into specific demographics like BIPOC or income-based groups. Among those that do provide racial data, for instance Pope et al.'s extensive cohort study from 2019, demonstrates that the hazard ratio (a potential marker of susceptibility) for all-cause mortality due to a unit increase in PM_{2.5} is 1.11 for non-Hispanic white, 1.15 for non-Hispanic black, and 1.20 for Hispanic populations nationwide. Yet, even this dataset has its limitations. Relying on nationwide data can overlook regional susceptibilities, which undoubtedly exhibit variations.

Thank you again for bringing up the topic of susceptibility. We believe it's worthwhile to include a brief discussion on this. We've added the following content to the fourth paragraph of our discussion section:

“Beyond exposure, it's important to recognize the cumulative impact where socio-economic, environmental, and health-related factors converge, increasing the susceptibility of DAC residents, especially the BIPOC population, to the adverse effects of TRAP. While our primary focus is on exposure, addressing the underlying determinants of this increased susceptibility can amplify the benefits of reducing TRAP exposure. To comprehensively evaluate the synergy between reducing exposure and susceptibility, future studies specifically focusing on susceptibility within DAC and BIPOC population are warranted.”

Reference: Pope, C. Arden, Lefler, Jacob S., Ezzati, Majid, Higbee, Joshua D., Marshall, Julian D., Kim, Sun Young, Bechle, Matthew, Gilliat, Kurtis S., Vernon, Spencer E., Robinson, Allen L., & Burnett, Richard T. Mortality risk and fine particulate air pollution in a large, representative cohort of U.S. adults. *Environ. Health Perspect.* **127**, (2019).

If that is the case, a philosophical question: what is the appropriate spatial scale at which to assess population disparities; near-road, census tract, city, county, or state? It seems the choice of domain could substantially influence the equity determination. This is also a potential limitation of working with a dispersion model, as opposed to a regional CTM. Some added discussion to this last point would be helpful, as would an analysis to determine the initial question asked in this paragraph.

Thank you for your comment. Both near-road and regional air quality (at the city, county, and state levels) are important. This is why the U.S. EPA maintains both near-roadway and ambient (which reflects regional) air quality monitoring stations. Our research aims to bridge the existing knowledge gap by establishing a framework for analyzing the near-roadway benefits of ZEVs using a bottom-up approach. Our work is intended to complement the Chemical Transport Model (CTM); and as you pointed out, the dispersion model has its limitations, such as being unable to model secondary pollutants.

As for the spatial resolution, there isn't a universally correct scale. Each spatial dimension, whether it's near-road, census tract, city, county, or state, offers a distinct vantage point and has its unique implications. We advocate for employing a bottom-up approach and refining the resolution as much as feasible. The finer the scale, the more detailed and reliable the analysis, especially from an Environmental Justice (EJ) standpoint where individual communities exhibit specific characteristics. You can then consolidate these data to broader spatial resolutions. In contrast, using a top-down approach often necessitates numerous assumptions, especially when trying to disaggregate. However, the bottom-up approach also demands extensive data collection and computational resources, which may not be feasible for every project. Therefore, adjustments should be made based on specific research question.

We have added the following to the third last paragraph in the Discussion section:

“Different climate mitigation policies could potentially lead to spatial heterogeneities in ambient air quality across communities. In terms of spatial resolution, each level of granularity, from near-road to the regional, provides distinct insights and implications. For more accurate environmental justice analyses, considering that individual communities have unique characteristics, it is beneficial to use the finest resolution possible. However, this granularity often requires extensive data collection and computational resources, which might not always be readily available. Thus, adjustments tailored to specific research projects may be needed.”

We have also added the limitation of the dispersion model in the Limitation subsection as follows:

“Moreover, our work aims to complement existing methodologies such as the CTM. It is important to note that the dispersion model has inherent limitations, including challenges in modeling secondary pollutants.”

Line 307: given battery weights and potential increases in brake and tire wear, do your PM estimates include greater PM emissions from these sources, or do you assume they are the same as the vehicles they replace? There’s been conflicting findings in the literature, so either method is justified, but the assumption should be noted.

Thank you for your insightful comment. We recognize that the emissions from brake and tire wear in ZEVs are on-going research topics, especially given the increased battery weights and regenerative braking in BEVs. In this study, our PM estimates, which include brake and tire wear, are sourced from the CARB EMFAC database. This database provides specific emission rates for different vehicle fuel technologies and weight classes.

Typo on Line 343 “re”.

Thank you for catching the typo. We have changed “re higher” to “are higher”.

For the ozone discussion on Lines 351-358, please refer to and cite Skipper et al (see their Figure 6).

Thank you for your comments. We have cited the Skipper et al.’s work. We have revised our manuscript as follows:

“This has been reported both in the LA100 study⁶⁰ conducted by the National Renewable Energy Laboratory and another recent study³³.”

Line 407-408: The authors may not have this data, but a question: from an air quality perspective, is it more effective to incentivize ZEV adoption in DACs or to incentivize reducing emission from or electrifying trucks? This question has real-world relevance to the IRA that prioritized passenger vehicle subsidies over reducing truck emissions.

Drawing from the year 2023 California EMFAC emission data as a reference, trucks (encompassing light-, medium-, and heavy-duty) emit 5.9 tons of PM_{2.5} per day, slightly less than the 6.0 tons emitted by passenger vehicles. In terms of NO_x emissions, trucks are responsible for 212.8 tons per day, which is considerably higher than the 92.5 tons from passenger vehicles. Thus, from an air quality standpoint alone, electrifying trucks should be prioritized.

However, the motivations behind the Inflation Reduction Act, which emphasizes subsidies for passenger vehicles, are likely multifaceted. Factors such as job creation,

bolstering local manufacturing, and political considerations may also play significant roles. Thus, for a more equitable transition, we proposed that subsidies be allocated both to promote ZEV adoption in DACs, considering their historical underfunding, and to accelerate truck electrification.

We have incorporated your suggestion and expand our last discussion paragraph to highlight the importance of making trucks zero-emission:

“Addressing the disparities inherent in environmental and health outcomes requires persistent and targeted efforts. As we move forward, future policies and incentive programs should take a comprehensive approach. Trucks, given their substantial emissions and frequent routes through DACs, pose a significant health risk. It is therefore important to prioritize the transition of trucks to zero-emission alternatives. In addition, addressing non-tailpipe emissions can provide transformative air quality improvements for the most vulnerable communities^{65,70,71}. By adopting this holistic approach, we are taking a decisive step towards achieving ZEV distributive justice and ensuring a just transition to clean transportation.”

REVIEWERS' COMMENTS

Reviewer #2 (Remarks to the Author):

The revisions and responses look good. Thank you for your attention to detail.